# Patterns of brain atrophy in recently-diagnosed relapsing-remitting multiple sclerosis

Rozanna Meijboom[1,2]*, Elizabeth N. York[1,2,3], Agniete Kampaite[1,2], Mathew A. Harris[4], Nicole White[1,2], Maria del C. Valdés Hernández[1,2], Michael J. Thrippleton[1,2], N. J. J. MacDougall[3], Peter Connick[1,3], David P. J. Hunt[1,5], Siddharthan Chandran[1,3], Adam D. Waldman[1,2]*, on behalf of the FutureMS Consortium[¶]

**1** Centre for Clinical Brain Sciences, University of Edinburgh, Edinburgh, United Kingdom, **2** Edinburgh Imaging, University of Edinburgh, Edinburgh, United Kingdom, **3** Anne Rowling Regenerative Neurology Clinic, University of Edinburgh, Edinburgh, United Kingdom, **4** Department of Psychology, University of Edinburgh, Edinburgh, United Kingdom, **5** UK Dementia Research Institute, University of Edinburgh, Edinburgh, United Kingdom

¶ Membership of the FutureMS Consortium is provided in the Acknowledgments
* Adam.Waldman@ed.ac.uk; rozanna.meijboom@ed.ac.uk

**Data Availability Statement:** The FutureMS project agreement, under reference LG/CPH/UOF001.2011, between The University Courts of the University of Glasgow, Edinburgh and

## Abstract

Recurrent neuroinflammation in relapsing-remitting MS (RRMS) is thought to lead to neurodegeneration, resulting in progressive disability. Repeated magnetic resonance imaging (MRI) of the brain provides non-invasive measures of atrophy over time, a key marker of neurodegeneration. This study investigates regional neurodegeneration of the brain in recently-diagnosed RRMS using volumetry and voxel-based morphometry (VBM). RRMS patients (N = 354) underwent 3T structural MRI <6 months after diagnosis and 1-year follow-up, as part of the Scottish multicentre 'FutureMS' study. MRI data were processed using FreeSurfer to derive volumetrics, and FSL for VBM (grey matter (GM) only), to establish regional patterns of change in GM and normal-appearing white matter (NAWM) over time throughout the brain. Volumetric analyses showed a decrease over time (q<0.05) in bilateral cortical GM and NAWM, cerebellar GM, brainstem, amygdala, basal ganglia, hippocampus, accumbens, thalamus and ventral diencephalon. Additionally, NAWM and GM volume decreased respectively in the following cortical regions, frontal: 14 out of 26 regions and 16/26; temporal: 18/18 and 15/18; parietal: 14/14 and 11/14; occipital: 7/8 and 8/8. Left GM and NAWM asymmetry was observed in the frontal lobe. GM VBM analysis showed three major clusters of decrease over time: 1) temporal and subcortical areas, 2) cerebellum, 3) anterior cingulum and supplementary motor cortex; and four smaller clusters within the occipital lobe. Widespread GM and NAWM atrophy was observed in this large recently-diagnosed RRMS cohort, particularly in the brainstem, cerebellar GM, and subcortical and occipital-temporal regions; indicative of neurodegeneration across tissue types, and in accord with limited previous studies in early disease. Volumetric and VBM results emphasise different features of longitudinal lobar and loco-regional change, however identify consistent atrophy patterns across individuals. Atrophy measures targeted to specific brain

Aberdeen, as well as the Grampian, Tayside, Lothian and Greater Glasgow Health boards, states that personal data cannot be released or made available to any third party unless the FutureMS steering committee have expressly permitted the data sharing, or personal data has been anonymized sufficiently. The magnetic resonance imaging data used in the current paper are not defaced and hence not sufficiently anonymized for open data sharing. Unrestricted access to these images in current form would present a confidentiality/privacy risk for participants. Additionally, the informed consent, approved by the South East Scotland Research Ethics Committee 02 under reference 15/SS/0233, states that by consenting to the FutureMS study, participants agree to sharing their data with the wider study team. The authors have not obtained explicit consent for deidentified data sharing beyond the wider study team. As such, they cannot make deidentified data openly available. The authors do, however, highly encourage collaboration with other research teams, and any researchers may request access to anonymized patient data from FutureMS-1 by contacting future-ms@ed.ac.uk. Proposals will be reviewed by the FutureMS steering committee and if approved, a signed data sharing agreement will be issued.

**Funding:** With thanks to FutureMS, hosted by Precision Medicine Scotland Innovation Centre (PMS-IC) and funded by a grant from the Chief Scientist Office, Scotland, to PMS-IC and Biogen Idec Ltd Insurance (combined funding under reference Exemplar SMS_IC010). Additional funding for authors came from the MS Society Edinburgh Centre for MS Research (grant reference 133; RM, AK), Chief Scientist Office – SPRINT MND/MS program (grant reference MMPP/01; ENY), Anne Rowling Regenerative Neurology Clinic (ENY), NHS Lothian Research and Development Office (MJT), the Row Fogo Charitable Trust (grant reference BROD.FID3668413; MVH), Wellcome Trust Senior Research Fellowship (grant reference 215621/Z/19/Z; DPJH), Medical Research Foundation (DPJH), and the UK Dementia Research Institute (SC), which receives its funding from UK DRI Ltd, funded by the UK Medical Research Council, Alzheimer's Society and Alzheimer's Research UK. Additional funding for the Edinburgh university 3T MRI Research scanner in RIE is funded by the Wellcome Trust (104916/Z/14/Z), Dunhill Trust (R380R/1114), Edinburgh and Lothians Health Foundation (2012/17), Muir Maxwell Research Fund, Edinburgh Imaging, and University of Edinburgh. The funders had no role in study

regions may provide improved markers of neurodegeneration, and potential future imaging stratifiers and endpoints for clinical decision making and therapeutic trials.

## 1. Introduction

Multiple sclerosis (MS) is a neuroinflammatory and neurodegenerative disease affecting two million people worldwide [1–3]. Disease progression and severity varies between individuals, and symptoms are diverse, including mobility and vision problems, pain, depression, fatigue and cognitive impairment [4]. Several MS subtypes have been identified, of which relapsing-remitting disease (RRMS) is the most common [5,6]. There is currently no cure for MS. More accurate biomarkers of disease progression and improved understanding of disease mechanisms, particularly in terms of neurodegeneration, are required for more suitable treatment of MS. Magnetic resonance imaging (MRI) allows for studying neurodegeneration *in vivo*, through measurement of tissue volume change (atrophy) over time, and may thus provide such valuable biomarkers of MS severity and progression.

White matter lesions (WMLs) on MRI, reflecting underlying inflammatory demyelinating lesions, are considered an imaging hallmark of RRMS and are required for RRMS diagnosis [7]. Modulating neuroinflammation is also the primary target for currently available disease modifying treatment (DMT) for RRMS. However, these appear to have only a limited effect on reducing associated neurodegenerative processes [8,9]. Previous studies have shown that neurodegeneration also plays a prominent role in the disease evolution of RRMS and is importantly already present in the early stages of the disease [10–15]. Early-stage grey matter (GM) atrophy has been observed in specific brain areas, including the cingulate gyrus, precuneus, thalamus, basal ganglia, brainstem and cerebellum [11,16–18]. Less is known about regional patterns of WM atrophy in RRMS, but WM atrophy also occurs in early disease and has been suggested to occur independently of WML development [10,19,20]. Furthermore, previous studies have shown microstructural changes reflective of neurodegeneration within the normal-appearing WM (NAWM) [21]. Importantly, atrophy appears to be a better predictor of clinical disability and deterioration than WMLs [13,17,22–32]. This suggests it could provide a useful surrogate for neurodegeneration, to help stratify patients and evaluate the efficacy of disease-slowing treatments. Existing studies investigating the effect of current DMTs on atrophy suggest a slowing of neurodegeneration after DMT use [33–38]. Further research is required to investigate the effect of current DMTs and possible future neuroprotective therapies on brain atrophy. Such studies would benefit from detailed knowledge of the location and extent of early-stage GM and WM atrophy in RRMS [39].

Current literature does not provide detailed insights into regional WM volume loss in recently-diagnosed RRMS, and only a limited number of studies have examined specific regional GM volume in early stages of the disease [11,16–18]. The aim of this study was to investigate neurodegenerative changes reflected in loco-regional atrophy in recently-diagnosed people with RRMS. Additionally, two analysis approaches were used to understand disparate results in the literature and the degree to which they result from different measurement methodology. Baseline and 1-year follow-up MR imaging data was used from the FutureMS study [40,41], a large multicentre cohort of RRMS patients within six months after diagnosis. This allows for identification of brain areas affected by early atrophy in RRMS, which may provide possible biomarkers of disease progression for DMT trials.

design, data collection and analysis, decision to publish, or preparation of the manuscript.

**Competing interests:** The authors have declared that no competing interests exist.

## 2. Methods

### 2.1 Participants

Participants with a recent diagnosis of RRMS according to the 2017 McDonald criteria ($< 6$ months) [7], were recruited (2016–2019) across five neurology sites in Scotland: Aberdeen, Dundee, Edinburgh, Inverness and Glasgow, as part of the FutureMS study [40,41]. Participants were 18 years or older and had the capacity to provide informed consent. Exclusion criteria were participation in a clinical trial prior to baseline assessment,contraindications for MRI, and intake of DMTs prescribed prior to baseline assessment. The latter criteria was selected because DMT intake in the Scottish population is very low compared with other countries, creating a unique opportunity to develop natural history predictive tools. Study visits took place at baseline (wave 0 [w0])) and 1-year follow-up (w1) and participants underwent brain MRI and expanded-disability status scale (EDSS) assessment at both time points. Full details of the FutureMS study have been previously described in Kearns et al. 2021 [40].

N = 431 participants underwent MR imaging at w0 and N = 382 participants underwent MR imaging at w1. The main reasons for not returning for w1 were not being able to reach participants at the provided contact details or participants having moved away from Scotland and not wanting to travel back. Additionally, the COVID-19 pandemic prematurely terminated w1 visits, which prevented several participants returning for their second visit. Additionally, 77/431 participants were excluded for various reasons (see Fig 1), resulting in N = 354 available MRI datasets for analysis.

All participants provided written informed consent before study entry. The study received ethical approval from the South East Scotland Research Ethics Committee 02 under reference 15/SS/0233 and was conducted in accordance with the Declaration of Helsinki and ICH guidelines on good clinical practice. Authors did not ave access to information that could identify individual participants during or after data collection. All data were anonymised with unique study identifiers.

### 2.2 MR image acquisition

MR image acquisition was performed across four sites in Scotland using comparable 3T MRI systems (Siemens in Glasgow, Dundee and Edinburgh; Philips in Aberdeen). Protocol harmonisation was implemented during the course of the study to increase between-site comparability and facilitate image analysis. Importantly, each participant underwent both MRI scans at the same centre using the same protocol. All participants underwent T1-weighted, T2-weighted, and 2D FLAIR imaging. Full details have been previously described in Meijboom et al. (2022) [41] (see S1 Table for an overview of all MR parameters).

### 2.3 MR image processing

Image processing methods have been previously described in full in Meijboom et al. (2022) [41] and are briefly summarised in the sections below.

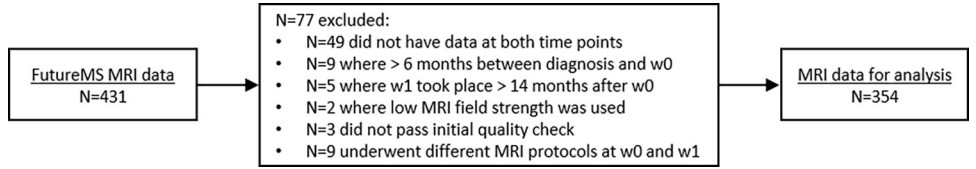

**Fig 1. Overview of data exclusions.**

**2.3.1 Registration and ICV.** All images were registered to the T1W image with a rigid body transformation (degrees of freedom = 6) using FSL FLIRT (FSL v6.0.1) [42,43] separately for each time point. Brain tissue was isolated using FSL BET2 (FSL v6.0.1) [44], followed by manual editing (i.e. removal of any non-intracranial tissue) of the resulting w0 intracranial image using ITK-SNAP v3.8.0 [45]. For each participant, the edited w0 intracranial image was registered to the w1 T1W image, to avoid within-subject variability between time points. Intracranial volume (ICV) was extracted using fslstats (FSL6.0.1).

**2.3.2 WML segmentation.** Hyperintense voxels on 2D FLAIR were identified by thresholding intensity values to 1.69 SDs > mean, using an adjusted method from Zhan et al. (2014) [46]. Resulting hyperintense areas unlikely to reflect pathology were removed using a predefined lesion distribution template [47], followed by Gaussian smoothing. Resulting WML masks were manually edited using ITK-SNAP [45]. WML volumes were extracted using fslstats and expressed as a ratio of ICV (r-ICV) to correct for head size.

**2.3.3 Tissue segmentation.** Tissue segmentation was performed on T1W images using the longitudinal processing stream [48] in FreeSurfer v6.0 (http://surfer.nmr.mgh.harvard.edu/) with default settings and the edited intracranial image as brain mask. Tissue segmentations were based on the Desikan-Kiliany atlas [49] and pial surfaces were improved using additional contrast from T2W images. Freesurfer output was manually edited by a trained neuroscientist where necessary using FreeView v2.0 (FreeSurfer v6.0) and following FreeSurfer editing guidelines (https://surfer.nmr.mgh.harvard.edu/fswiki/Edits). WML masks were then subtracted from the resulting tissue segmentations using fslmaths (FSL6.0.1), to create final tissue masks. Global and regional tissue volumes (see Table 1) were calculated from the final tissue masks using fslstats, and subsequently expressed as r-ICV to correct for head size.

**2.3.4 Voxel-based morphometry.** Longitudinal voxel-based morphometry (VBM), applying a voxel-wise comparison of local GM concentration between w0 and w1, was performed using the FSL VBM pipeline (FSL6.0.1; http://fsl.fmrib.ox.ac.uk/fsl/fslwiki/FSLVBM) [50–53].

**Table 1. Global and regional GM/WM and subcortical areas.**

| Global WM and GM | Brainstem<br>Cortical GM<br>Cerebellar GM<br>Cerebral WM<br>Cerebellar WM | | | Subcortical regions | Amygdala<br>Basal ganglia<br>Hippocampus<br>Nucleus accumbens<br>Thalamus<br>Ventral diencephalon |
|---|---|---|---|---|---|
| Regional WM and GM | *Frontal* | *Parietal* | *Temporal* | *Occipital* | |
| | Superior frontal | Superior parietal | Superior temporal | Lateral Occipital | Insula |
| | Caudal middle frontal | Inferior parietal | Middle temporal | Lingual | |
| | Rostral middle frontal | Supramarginal | Inferior temporal | Cuneus | |
| | Pars opercularis | Postcentral | Superior temporal sulcus banks | Pericalcarine | |
| | Pars triangularis | Precuneus | Fusiform | | |
| | Pars orbitalis | Posterior cingulate | Transverse temporal | | |
| | Lateral orbitofrontal | Isthmus cingulate | Entorhinal | | |
| | Medial orbitofrontal | | Temporal Pole | | |
| | Precentral | | Parahippocampal | | |
| | Paracentral | | | | |
| | Frontal pole | | | | |
| | Caudal anterior cingulate | | | | |
| | Rostral anterior cingulate | | | | |

Left and right hemisphere were separately included for all areas, excluding the brainstem. WM = white matter, GM = grey matter.

It was performed for GM only, because GM VBM is a more commonly used and accepted method (e.g. a WM VBM pipeline is not implemented in FSL). The first step of the VBM pipeline (i.e. brain extraction) was omitted, as we were able to use the previously created intracranial brain images, as described above. In step two, GM, WM and CSF were segmented from T1W images and the study-specific GM template was created. Specifically, all subject GM images at both time points were affine-registered to the GM ICBM-152 template, and then concatenated and averaged to create a study-specific GM template. The subject GM images were then non-linearly registered to this study-specific template, and concatenated and averaged again to then create a final study-specific GM template in standard space ($2 \times 2 \times 2mm^3$ resolution). In the last step, subject GM at both time points was then non-linearly registered to this study-specific template, after which a Jacobian modulation was applied, resulting in modulated GM images for each time point per subject.

As an additional step, to allow for longitudinal VBM analysis, we subtracted the modulated GM images for w1 from those at w0 using fslmaths, resulting in a GM difference file for each subject. Modulated GM difference files were then smoothed with a Gaussian size 2 kernel using fslmaths, and concatenated using fslmerge (FSL6.0.1) to form a final 4D group GM difference image for statistical analysis.

## 2.4 Statistical analyses

**2.4.1 Volumetrics.** Statistical analysis for tissue volumes was performed using R v4.0.2 [54] and package lmerTest [55], with further packages ggplot2 [56] and lattice [57] used to create figures. A linear mixed-effects model was applied for each tissue volume (Table 1) with time point (w0, w1) as regressor of interest and age, sex, imaging site, WML change (w1 – w0) and DMT status at w1 as covariates. WML volume change was added as a covariate to correct for GM and WM volume changes induced by enlarging and/or receding WMLs. DMT status was included as a covariate to correct for the difference in effect of undergoing or not undergoing DMT (between baseline and follow-up) on GM and WM volume change. Additionally, to correct for any differences in volume change over time due to older and younger participants, the interaction between time and age was added. The interaction was removed from the model if not significant. For each outcome variable, extreme outliers (>3 SD) as well as participants with missing data at either time point were excluded. In case of missing DMT data, the mode of the group was used instead, i.e. yes to DMT use at w1. Subsequently, continuous variables were scaled before being entered into the linear model. False discovery rate (FDR) correction was performed to adjust p-values for multiple comparisons, with corrected p-values (q-values) considered significant at q<0.05.

**2.4.2 Voxel-based morphometry.** FSL Randomise (FSL v6.0.1) [58] was used to perform nonparametric permutation analysis on the VBM output. A general linear model (GLM) design matrix was created with GM difference, age, sex, imaging site, DMT status at w1 and subject-specific WML mask as explanatory variables (EV). In case of missing DMT data, the mode of the group was used instead, i.e. yes to DMT use at w1. The interaction of age and time was not added here. Given the interaction was not significant in 98% of volumetric regions studied, it is unlikely it will have a significant effect on the overall VBM results. EVs were recentered to zero where appropriate.

WMLs were included as EVs to ensure their exclusion from the GM images. Subject-specific WML EVs were created by a) registering all subject-specific WML masks at w0 and w1 to the subject's modulated GM segmentation image at respectively w0 and w1, with the VBM transformation matrix that was created for registration of the native T1W to the group-specific template (fslmaths), and b) using FSL tool 'setup_masks' (FSL6.0.1; https://fsl.fmrib.ox.ac.uk/

fsl/fslwiki/Randomise/UserGuide) to generate a concatenated 4D subject-specific WML image from the step a output images, as well as add a WML EV for each subject separately to the design matrix.

One t-contrast was defined, which assessed GM change over time, corrected for the remaining EV's. The 'randomise' function was run using 5000 permutations, and with threshold-free cluster enhancing (TFCE) [59] to perform a voxel-wise analysis with a specific focus on voxel clusters. Cluster (FSL6.0.1) was used to extract resulting clusters and atlasquery (FSL v6.0.1) was used to obtain labels for the included anatomical regions. Results were family-wise error (FWR) corrected for multiple comparisons (p<0.001).

## 3. Results

### 3.1 Demographics

Data from N = 315 participants were used for tissue volume analysis and data from N = 351 were used for VBM analysis. Specifically, for tissue volume analysis, 39/354 cases were excluded due to segmentations of insufficient quality and for VBM analysis, 3/354 datasets were excluded for data processing purposes. In addition, for the tissue volume dataset, for participants where segmentation failed for one or a small number of areas only (usually subcortical), the appropriate areas were excluded from analysis, but the remaining data of the participant was included in analysis. See Table 2 for participant demographics for each analysis type.

Participants were DMT-naïve at w0, but most started receiving DMTs between w0 and w1 (Table 2). Post-hoc regression analyses showed tissue volume change was not significantly different between participants positive or negative for DMTs at w1 (S2 Table). The majority of DMTs administered to this cohort consisted of dimethyl fumarate (97 volumetrics sample; 104 VBM sample), but glatirimer acetate (31; 37), beta interferon (16; 20) and alemtuzumab (22; 24) were also used. A few participants received natalizumab (8 volumetrics sample; 9 VBM sample), fingolimod (7; 7), teriflunomide (5; 9), azathioprine (1; 1), a combination of the

**Table 2. Participant demographics.**

|  | **Tissue volumes** | **VBM** |
|---|---|---|
| **N (fenale)** | 315 (237) | 351 (264) |
| **Mean age at w0 (SD)** | 38.18 (10.21) | 38.22 (10.38) |
| **Median EDSS at w0 (IQR)** | 2 (1.5) | 2 (1.5) |
| **Median EDSS at w1 (IQR)** | 2.5 (1) | 2.5 (1) |
| **Mean diagnosis to w0 in days (SD)** | 63.77 (39.12) | 64.02 (39.04) |
| **DMT use** | w0: none \| w1: 224 (8 unknown) | w0: none \| w1: 252 (unknown 13) |
| **MRI sites** | ABN = 15, ED1 = 74, ED2 = 71, DUN = 34,GLA = 121 | ABN = 17, ED1 = 84, ED2 = 77, DUN = 38,GLA = 135 |
| **MRI protocol** | protocol A = 134, protocol B = 181 | protocol A = 149, protocol B = 202 |
| **Mean WML volume %ICV at w0 (SD)** | 0.82 (0.67) | 0.91 (0.89) |
| **Mean WML volume %ICV at w1 (SD)** | 1.00 (0.66) | 1.09 (0.84) |

N = sample size, SD = standard deviation, IQR = interquartile range, EDSS = expanded disability status scale, VBM = voxel-based morphometry, w0 = baseline, w1 = 1-year follow-up, MRI = magnetic resonance imaging, WML = white matter lesion, ICV = intracranial volume, ABN = Aberdeen, ED1 = Edinburgh site 1, ED2 = Edinburgh site 2, DUN = Dundee, GLA = Glasgow.

aforementioned DMTs (16; 18) or a DMT not further specified (21; 24). Additionally, 9 participants received a short course of oral or intravenous steroids in keeping with UK guidelines for managing MS relapse [60], within six weeks prior to the MRI scan (range 3 to 31 days; 7/9 prior to w0 and 2/9 prior to w1).

## 3.2 Global GM/WM, subcortical and whole-brain volumes

Whole-brain, brainstem and left and right cerebral GM and WM, right cerebellar GM, brainstem, amygdala, basal ganglia, hippocampus, nucleus accumbens, thalamus and ventral diencephalon volumes significantly (q<0.05) decreased over time, whereas left and right cerebellar WM did *not* significantly (q>0.05) change over time (Table 3, Fig 2). Additionally, atrophy rates for the left cerebellar GM were significantly different across age (See S3 Table for interaction effects). Covariate significance (p<0.05) was variable across regions. Age and sex were significant in nearly all regions, except the bilateral cerebral NAWM (age and sex), brainstem (sex), right cerebellar GM (sex), and bilateral cerebellar NAWM (age).Site was significant for the bilateral amygdala, left accumbens, right cerebellar GM, bilateral cerebral GM, bilateral hippocampus, bilateral cerebellar NAWM, left cerebral NAWM, left thalamus (site), and

**Table 3. Global GM/NAWM, subcortical and whole-brain volumes results for change over time.**

| | $B_{standardised}$ | SE | df | t | $P_{uncorrected}$ | CI 2.5. | CI 97.5. | Mean w0 (%ICV) | SD w0 (%ICV) | Mean w1 (%ICV) | SD w1 (%ICV) | Mean 1-year % change |
|---|---|---|---|---|---|---|---|---|---|---|---|---|
| Accumbens L | -0.0546 | 0.0210 | 313 | -2.6078 | **0.0096** | -0.0958 | -0.0135 | 0.0371 | 0.0070 | 0.0367 | 0.0071 | -1.0328 |
| Accumbens R | -0.0787 | 0.0229 | 288 | -3.4413 | **0.0007** | -0.1236 | -0.0338 | 0.0407 | 0.0063 | 0.0403 | 0.0064 | -0.8147 |
| Amygdala L | -0.0963 | 0.0222 | 312 | -4.3368 | **<0.0001** | -0.1398 | -0.0527 | 0.1018 | 0.0119 | 0.1007 | 0.0120 | -1.1297 |
| Amygdala R | -0.1458 | 0.0217 | 313 | -6.7159 | **<0.0001** | -0.1885 | -0.1032 | 0.1156 | 0.0117 | 0.1139 | 0.0116 | -1.4730 |
| Basal Ganglia L | -0.1510 | 0.0192 | 313 | -7.8818 | **<0.0001** | -0.1887 | -0.1134 | 0.6456 | 0.0826 | 0.6359 | 0.0832 | -1.5068 |
| Basal Ganglia R | -0.1646 | 0.0200 | 313 | -8.2182 | **<0.0001** | -0.2039 | -0.1253 | 0.6477 | 0.0779 | 0.6362 | 0.0761 | -1.7785 |
| Brainstem | -0.0671 | 0.0136 | 312 | -4.9164 | **<0.0001** | -0.0938 | -0.0403 | 1.3243 | 0.1193 | 1.3165 | 0.1196 | -0.5904 |
| GM cerebellar L[a] | -0.0520 | 0.0115 | 313 | -4.5135 | **<0.0001** | -0.0745 | -0.0294 | 3.4160 | 0.3488 | 3.3981 | 0.3459 | -0.5247 |
| GM cerebellar R | -0.0615 | 0.0127 | 313 | -4.8253 | **<0.0001** | -0.0865 | -0.0365 | 3.4578 | 0.3922 | 3.4367 | 0.3888 | -0.6097 |
| GM cortical L | -0.0861 | 0.0165 | 313 | -5.2088 | **<0.0001** | -0.1186 | -0.0537 | 15.5401 | 1.0006 | 15.4544 | 0.9889 | -0.5513 |
| GM cortical R | -0.0608 | 0.0169 | 313 | -3.5949 | **0.0004** | -0.0941 | -0.0276 | 15.5598 | 0.9803 | 15.5002 | 0.9790 | -0.3829 |
| Hippocampus L | -0.1005 | 0.0177 | 313 | -5.6859 | **<0.0001** | -0.1352 | -0.0658 | 0.2669 | 0.0254 | 0.2645 | 0.0255 | -0.8973 |
| Hippocampus R | -0.0951 | 0.0163 | 313 | -5.8384 | **<0.0001** | -0.1271 | -0.0632 | 0.2784 | 0.0673 | 0.2762 | 0.0696 | -0.8018 |
| NAWM cerebellar L | -0.0215 | 0.0168 | 313 | -1.2787 | 0.2019 | -0.0546 | 0.0115 | 0.9693 | 0.5568 | 0.9658 | 0.5386 | -0.3551 |
| NAWM cerebellar R | 0.0096 | 0.0162 | 311 | 0.5956 | 0.5519 | -0.0221 | 0.0414 | 0.9165 | 0.1113 | 0.9169 | 0.1098 | 0.0444 |
| NAWM cerebral L | -0.1378 | 0.0121 | 311 | -11.3639 | **<0.0001** | -0.1617 | -0.1140 | 14.0731 | 1.1201 | 13.9223 | 1.1618 | -1.0716 |
| NAWM cerebral R | -0.1263 | 0.0120 | 313 | -10.5151 | **<0.0001** | -0.1498 | -0.1027 | 14.0646 | 1.0971 | 13.9226 | 1.1497 | -1.0100 |
| Thalamus L | -0.1163 | 0.0116 | 313 | -10.0684 | **<0.0001** | -0.1390 | -0.0937 | 0.4972 | 0.7444 | 0.4908 | 0.7296 | -1.2765 |
| Thalamus R | -0.1055 | 0.0107 | 311 | -9.8826 | **<0.0001** | -0.1265 | -0.0846 | 0.4456 | 0.0441 | 0.4409 | 0.0450 | -1.0525 |
| Ventral DC L | -0.0763 | 0.0173 | 311 | -4.4036 | **<0.0001** | -0.1103 | -0.0423 | 0.2568 | 0.0230 | 0.2550 | 0.0231 | -0.6912 |
| Ventral DC R | -0.1306 | 0.0186 | 289 | -7.0225 | **<0.0001** | -0.1671 | -0.0941 | 0.2587 | 0.0246 | 0.2560 | 0.0245 | -1.0304 |
| Whole-brain | -0.1001 | 0.0146 | 313 | -6.8463 | **<0.0001** | -0.1288 | -0.0714 | 74.1152 | 3.7355 | 73.7433 | 3.7808 | -0.5018 |

Global GM/NAWM, subcortical and whole-brain volumes results for change over time (w1-w0) as assessed with a linear mixed-effects model, corrected for age, sex, imaging site, DMT status at w1 and WML change. Regression coefficient shown are standardised. Mean and SD are shown for volumes as % of intracranial volume (% ICV), without covariate correction. B = standardised beta value, SE = standard error, df = degrees of freedom, CI = confidence interval for beta value, SD = standard deviation, w0 = baseline, w1 = 1-year follow-up, L = left, R = right, GM = grey matter, NAWM = normal-appearing white matter, WML = white matter lesion, DC = diencephalon, ICV = intracranial volumep-values surviving FDR correction (q<0.05) are highlighted in bold.
[a]The model for GM cerebellar L included the significant interaction term for time*age.See S3 Table.

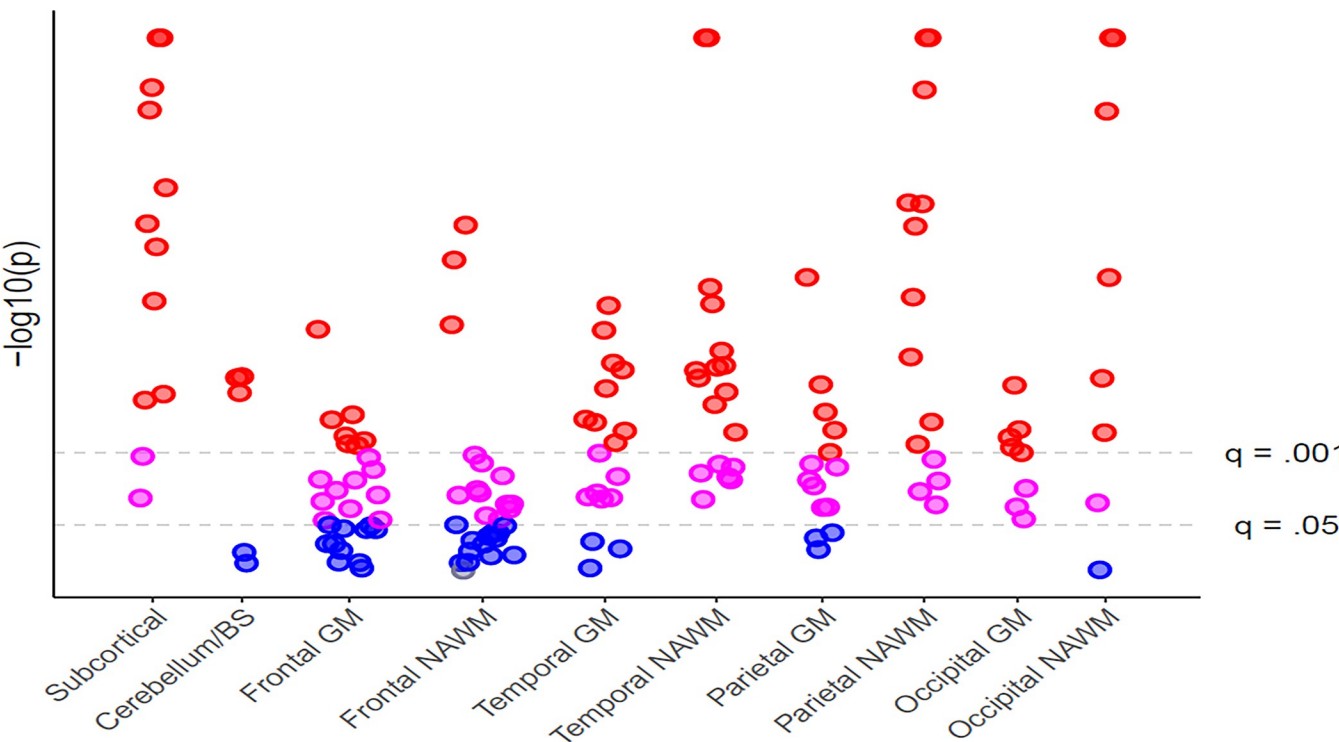

**Fig 2. Manhattan plot of longitudinal brain volume change.** Each point represents a tissue region within the given brain area category (x-axis). P-values were inverse log transformed (y-axis). Insignificant p-values (p>0.05) are shown in blue. Significant (p<0.05) false discovery rate (FDR) corrected p-values at q = 0.05 are shown in pink, and in red for q = 0.001. Regardless of significance, the direction of all effects – but one frontal NAWM area (in grey)—showed a longitudinal volume decrease. BS = brainstem, GM=grey matter, NAWM = normal-appearing white matter.

whole-brain (site). WML change was significant for the bilateral amygdala and right thalamus only. DMT status at w1 did not reach significance for any brain region. See S3 Table for an overview of all significant covariates and interaction effects.

### 3.3 Regional cortical GM and cerebral WM

Regional GM and NAWM volume decrease (q<0.05) was observed in most regions and predominantly in the temporal, parietal and occipital lobe. GM volume decreased over time in 15/18 temporal regions, 16/26 frontal regions, 11/14 parietal regions, 8/8 occipital regions and the bilateral insula. NAWM volume decreased over time in 18/18 temporal regions, 14/26 frontal regions, 14/14 parietal regions, 7/8 occipital regions and the insula (L). More GM and NAWM regions showed volume loss in the left than in the right hemisphere, particularly in the frontal lobe. See Figs 2 and 3, and S4 Table. Covariate significance was variable across regions, but age was most consistently significant and sex as well as site to a lesser extent. In contrast, WML change and DMT status at w1 only reached significance for very few brain regions. Additionally, 2/14 frontal NAWM regions, i.e. the left and right superior frontal NAWM, showed differential atrophy rates across age. See S3 Table for an overview of all significant covariates and interactions.

### 3.4 Voxel-based GM change

Twelve clusters of various sizes (Table 4) showed a change in local concentration of GM over time. The largest clusters were centred within 1) the temporal lobe and subcortical areas, 2)

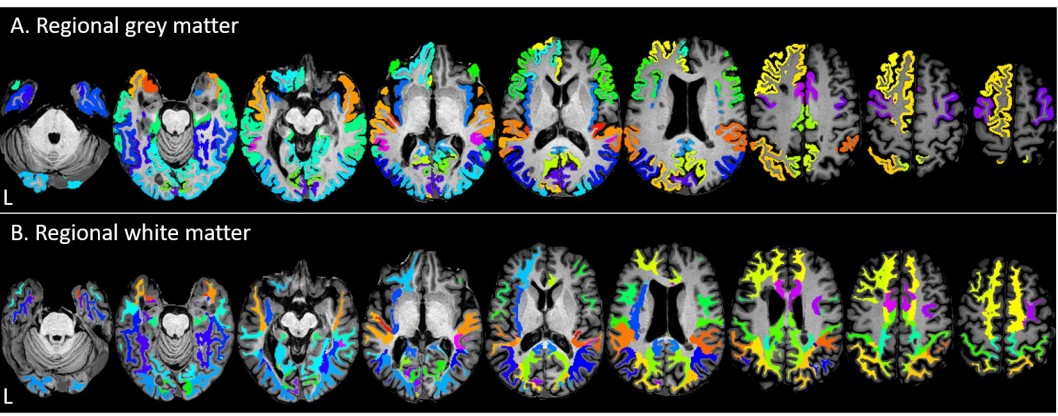

**Fig 3. Volumetric results.** For illustration purposes, regional grey matter (GM) (A) and normal-appearing white matter (NAWM) (B) areas with significant volume decrease (w1-w0; q<0.05) are shown on an example subject's axial T1-weighted image. Colours were chosen to emphasise borders between regions and have no further meaning in terms of results. This figure was created using MRIcron (https://www.nitrc.org/projects/mricron).

cerebellum, 3) anterior cingulum and supplementary motor cortex. Additional smaller clusters were also observed in the temporal lobe and cerebellum. Furthermore, four smaller clusters were observed in the occipital lobe, as well as one cluster in the posterior frontal lobe and one in the parietal supramarginal and angular gyrus. See Fig 4 for VBM results. See Fig 5 for a visual comparison of volumetric (regional GM, cerebellar GM and subcortical areas) and VBM results.

**Table 4. Voxel-based morphometry results.**

| Cluster Index | Voxels | MAX X (mm) | MAX Y (mm) | MAX Z (mm) | COG X (mm) | COG Y (mm) | COG Z (mm) | MAX anatomical location | COG anatomical location |
|---|---|---|---|---|---|---|---|---|---|
| 1 | 7415 | 34 | 4 | -44 | 3.03 | -1.47 | -4.59 | Temporal Pole, Anterior Temporal Fusiform Gyrus, Anterior Inferior Temporal Gyrus | Right thalamus |
| 2 | 2882 | 20 | -64 | -52 | 1.76 | -65.2 | -32.1 | Cerebellum | Cerebellum |
| 3 | 794 | 0 | 22 | 20 | -0.111 | -1.98 | 45.5 | Anterior Cingulate Gyrus | Anterior Cingulate Gyrus, Supplementary Motor Cortex |
| 4 | 83 | 42 | 12 | 26 | 42.1 | 9.71 | 30.8 | Inferior/Middle Frontal Gyrus, Precentral Gyrus | Inferior/Middle Frontal Gyrus, Precentral Gyrus |
| 5 | 74 | -24 | -76 | -12 | -22.8 | -73.9 | -11.4 | Occipital Fusiform Gyrus, Lateral Occipital Gyrus, Lingual Gyrus | Occipital Fusiform Gyrus, Lingual Gyrus |
| 6 | 62 | 34 | -96 | -10 | 27.1 | -97.8 | -5.35 | Occipital Pole, lateral occipital cortex | Occipital Pole |
| 7 | 39 | 14 | -98 | 18 | 10.7 | -95.4 | 22.9 | Occipital Pole | Occipital Pole |
| 8 | 38 | 60 | -48 | 18 | 55.7 | -47.2 | 20.9 | Angular Gyrus, Posterior Supramarginal Gyrus | Angular Gyrus, Posterior Supramarginal Gyrus |
| 9 | 37 | -44 | -20 | -36 | -45.9 | -19.1 | -31.5 | Posterior Inferior Temporal Gyrus, Posterior Temporal Fusiform Gyrus | Posterior Inferior Temporal Gyrus, Posterior Temporal Fusiform Gyrus |
| 10 | 32 | 20 | -102 | 4 | 16.8 | -101 | 7.44 | Occipital Pole | Occipital Pole |
| 11 | 27 | -62 | -24 | 4 | -63.3 | -23.6 | 8.96 | Posterior Superior Temporal Gyrus, Planum Temporale | Posterior Superior Temporal Gyrus, Planum Temporale |
| 12 | 26 | -32 | -44 | -42 | -31.6 | -40.5 | -40.9 | Cerebellum | Cerebellum |

VBM cluster results (cluster-size >20, $p_{corrected}$<0.001) for GM change over time (w1-w0) in RRMS (N = 351). VBM = voxel-based morphometry, GM = grey matter, RRMS = relapsing-remitting multiple sclerosis, MAX X/Y/Z = maximum cluster coordinates, mm = millimetres, COG X/Y/Z = centre of gravity cluster coordinates.

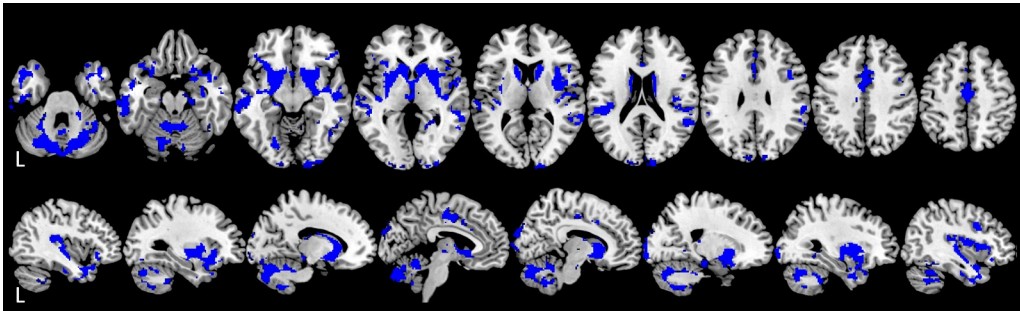

**Fig 4. Voxel-based morphometry (VBM) results.** VBM results for significant grey matter (GM) change over time (w1-w0; p_corrected<0.001; N = 351) in RRMS are shown in blue on a template axial (top row) and sagittal (bottom row) T1-weighted image. This figure was created using MRIcron (https://www.nitrc.org/projects/mricron) and the implemented 'ch2' template (http://www.bic.mni.mcgill.ca/ServicesAtlases/Colin27).

## 4. Discussion

The current study used imaging data from the FutureMS cohort, a large longitudinal multicentre cohort of people with recently-diagnosed RRMS in Scotland. The aim of our study was to establish a profile of regional NAWM and GM atrophy in recently-diagnosed RRMS. In addition, the effects of different imaging analysis approaches on observed patterns of atrophy were compared.

Widespread NAWM and GM atrophy was observed using both approaches. With the volumetric approach, people with recently-diagnosed RRMS showed brain volume loss in the brainstem, all subcortical regions and the cerebellar GM but not NAWM, over one year. Furthermore, cerebral NAWM and GM volume loss over one year was also evident. Specifically, GM and NAWM volume decreased in nearly all temporal, parietal and occipital regions and in about half of frontal regions. Slightly more NAWM regions showed volume decrease in the temporal-parietal lobes and slightly more GM regions showed a volume decrease in the frontal lobe. Additionally, more *left* frontal GM and NAWM regions showed volume decrease than *right* frontal regions. With the VBM approach (GM only), most of these subcortical and cortical GM findings in recently-diagnosed RRMS were replicated. Large areas of change in GM concentration were observed in temporal and subcortical regions, as well as in the cerebellum and the anterior cingulum, with some smaller areas in the occipital lobe. However, overall,

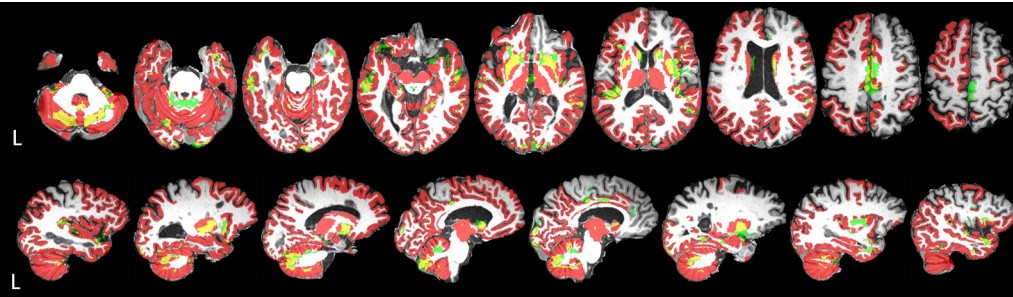

**Fig 5. Comparison of voxel-based morphometry (VBM) and volumetric results in relapsing-remitting multiple sclerosis (RRMS).** For illustration purposes, VBM results for significant grey matter (GM) change over time (w1-w0; p_corrected<0.001) (green) are overlaid on volumetric results (red) for regional GM, cerebellar GM and subcortical areas with significant volume decrease (w1-w0; q<0.05) (red). Areas where VBM and volumetry overlap are shown in yellow. Overlays are shown on an example subject's axial (top row) and sagittal (bottom row) T1-weighted image. This figure was created using MRIcron (https://www.nitrc.org/projects/mricron).

fewer regions of GM change were observed than with the volumetric approach, particularly in the parietal and frontal lobes, without prominent hemispheric differences.

## 4.1 Subcortical atrophy

The importance of subcortical neurodegeneration in MS has been recently summarised by Ontaneda et al. (2021), emphasising that atrophy of deep GM structures is common in MS and may be a suitable target for DMT treatment [61]. In this study, we observed prominent subcortical volume changes as well as a large cluster of GM concentration changes within subcortical regions in early RRMS. These result are in line with previous studies reporting early-stage subcortical atrophy, specifically in the thalamus and basal ganglia [11,16,22]. The thalamus has been suggested to play an important role in MS disease symptomatology, with lower volumes being predictive of clinical worsening [62], even in the early stages [63]. Basal ganglia changes have also frequently been observed and related to clinical change [64,65]. Additionally, our study showed that other subcortical regions, such as the amygdala, hippocampus, ventral diencephalon and nucleus accumbens, also show early-stage atrophy. This is in line with previous studies reporting similar subcortical volume changes as well as associations with clinical changes in clinical isolated syndrome (CIS) or later stages of RRMS [65–72].

## 4.2 Cerebellar and brainstem atrophy

The observed GM volume changes are in line with previous reports of early-stage cerebellar atrophy [16] and they are unsurprising as the cerebellum has been shown to play an essential role in both sensorimotor and cognitive dysfunction in MS [73–76]. VBM analysis also indicated a large cluster of GM changes within the cerebellum. Cerebellar NAWM change was not observed, which is in line with previous studies reporting absence of cerebellar NAWM volume change in CIS and RRMS [77,78]. In contrast, some studies do report cerebellar NAWM atrophy in MS [79,80], however this may be explained by different disease stages being studied as well as study samples lacking a distinction between MS subtypes. Moreover, studies using diffusion tensor imaging (DTI) to explore cerebellar changes in RRMS have observed early-stage NAWM microstructural changes in absence of macrostructural NAWM volume change [81]. This suggests that microstructural cerebellar damage is already present in early-stage RRMS, which may go on to develop into macrostructural cerebellar NAWM atrophy in later stages.

The brainstem is also known to be involved in MS, showing both evidence of WMLs as well as volume loss, which are associated with clinical symptomatology [82–84]. Here we observed brainstem volume loss in recently-diagnosed RRMS, which is corroborated by findings from Eshaghi et al. (2018) [11]. Overall, this may suggest that atrophy of both the brainstem and cerebellar GM plays an important role in early-stage neurodegeneration in RRMS.

## 4.3 Cerebral regional NAWM and GM atrophy

This study suggests that both regional NAWM and GM are evident in recently-diagnosed RRMS in all brain lobes in variable degrees. Using volumetrics, cerebral GM volume loss was observed more prominently in the temporal, parietal and occipital lobes than in the frontal lobe. Similarly, with VBM, prominent GM changes were observed in the temporal, occipital and posterior frontal lobes, but less so in the anterior frontal lobe and parietal lobe. This discrepancy may be explained by the changes within the latter regions being less localised which may have left them undetected by VBM. The early GM changes observed here are partly in line with previous studies showing prominent early changes in the occipital [16], temporal [17,72,85], parietal (posterior cingulum) [11] and frontal lobe [85]. However, these

observations differ from the current results in that they do not report on GM changes in all lobes simultaneously. This is likely explained by differences in methodology [86] (e.g. processing methods, inclusion of healthy controls, design), which is also supported by the discrepancy shown between VBM and volumetric results within this study.

Cerebral NAWM loss has been observed in previous studies [10,19,20], but to our knowledge, regional NAWM atrophy has not been investigated before. Similar to regional GM change, we observed widespread cerebral NAWM loss, independent of lesion accumulation, which was most prominent in the temporal, occipital and parietal regions. This suggests that NAWM neurodegeneration is also evident in recently-diagnosed RRMS. In comparison with cerebral GM volume loss, cerebral NAWM volume loss was observed in slightly more regions in temporal and parietal lobes, but not in the frontal lobe where more GM regions showed volume loss. This may support the notion that pathological processes underlying GM and NAWM change are at least partly dissociated [87].

More frontal NAWM and GM regions were affected in the left than in the right hemisphere - using the volumetric approach - whereas the temporal, parietal and occipital lobes were symmetrically affected. A left-hemispheric predilection for atrophy in all brain lobes has been observed previously in MS, although only in a small number of studies [88–90]. Moreover, conflicting results have been reported in studies investigating hemispheric asymmetry in damage accumulation, in both ageing and neurological disease [88]. Overall, it remains unclear whether asymmetry of neurodegeneration is a feature of MS, and whether the degree of differentially-affected cognitive functions is associated with the left and right frontal hemispheres; further research is required to elucidate this.

Surprisingly, we did not detect volumetric changes within areas in the frontal lobe associated with sensorimotor functioning, typically impaired in MS, such as in the precentral GM/NAWM and paracentral GM. However, paracentral NAWM volumetric change was indeed observed and the VBM results were also indicative of a small cluster of GM change within the supplemental motor area and precentral gyrus. Previous literature has shown GM volume change in the precentral gyrus in early RRMS compared with healthy controls [91], as well as abnormal iron deposition in the precentral gyrus in later stage MS [92]. This suggests that motor cortex abnormalities are involved in MS as expected, however the lack of volumetric precentral GM changes observed in our study may indicate that widespread motor cortex change over time may not be as fast as in other frontal regions in early RRMS.

### 4.4 Volumetry and VBM

The findings of this study emphasise that different analysis approaches can lead to significantly different results from the same data. Although GM change was observed in overlapping areas, there were key differences between the VBM and volumetric output. A volumetric approach like FreeSurfer segments brain regions, allowing calculation of the total number voxels within these regions. VBM on the other hand does not require regions to be pre-defined and looks for changes voxel-by-voxel, allowing for detection of local and possibly smaller changes [53]. Furthermore, FSL VBM also uses TFCE, which enhances clusters of voxels that show change increasing focus on spatially-localised changes, even when these changes are small in overall magnitude [59]. Conversely volumetry may not detect such subtle localised change, but it is more sensitive to distributed change across a region, which would be likely missed by VBM.

Further differences in processing steps and statistical methods are also likely to cause outcome differences. For example, VBM treats all data as one group and registers all participants to a common template, which may lead to some individual change being lost. FreeSurfer avoids this by processing participant data separately in their own native space. However,

between-subject cortical variability can affect accuracy of atlas-based approaches and may thus bias the volumetric results [93]. Another example is that FSL VBM uses FWE correction for multiple comparisons, whereas we have applied FDR for the volumetric analysis. As FWE is considered to be more stringent than FDR, it is possible that VBM underestimated and FDR overestimated actual change, leading to a difference in results [94]. Additionally, a larger number of data failed quality checks with FreeSurfer than with VBM, leading to a difference in sample size between the two methods. Although this may suggest that VBM is a more robust method in case of lower data quality, alternatively, this may also indicate that FreeSurfer has a better and stricter quality-check procedure in place.

Specifically, for our RRMS cohort, the difference in results may indicate that the frontal and parietal changes observed with the volumetric approach are more widespread and more variable across participants. On the other hand, the overlapping temporal, occipital and subcortical results, may be indicative of prominent changes within these regions. Overall, both VBM and volumetric approaches have advantages and important limitations that should be taken into consideration when interpreting results.

### 4.5 Limitations

This study has some limitations. First, FutureMS is a multi-centre study involving five different MR systems and two different (although very similar) MR protocols, which may have influenced the results. The parameters for the T1W acquisition, upon which the current analyses were mainly based are, however, nearly identical between protocols. Additionally, all participants underwent their MR assessment with the same system and protocol at both time points. Pooling of data across centres is important for clinical studies, as it allows for drawing conclusions based on larger datasets. Second, we have not studied the relationship between atrophy and clinical features in this study. The reason for this is that divergence in clinical disease course is limited over one year in early MS and may be confounded by recovery from acute inflammatory episodes that led to initial diagnosis. We are therefore initially concentrating on patterns of atrophy between the point of diagnosis and one year and will focus our future research on clinical correlates of atrophy as they develop over a longer timescale. Third, pseudoatrophy may have been caused by spontaneous recovery from inflammation and associated swelling. This cannot be corrected for, and should be taken into consideration when interpreting results. Pseudoatrophy may also have been caused by DMTs, which we have carefully corrected for by including adding DMT as a covariate. Results showed that DMTs did not have a significant effect on whole-brain, cerebral, cerebellar, brainstem or subcortical atrophy, and only in 6 out of 105 WM/GM regions that showed significant atrophy, which was corrected for accordingly. Finally, matched healthy control data were not available for comparison, which may also be considered a limitation. It is worth noting, however, that overall whole-brain volume decrease observed in this study was 0.5%; this exceeds brain atrophy expected to occur in healthy individuals within the same age range (30-40 years) [95], as well as the atrophy threshold (0.4%) proposed in the 2020 MAGNIMS guidelines [39]. Importantly, the longitudinal study design mitigates significant variations in brain volume between individuals, and allows regional and global atrophy to be mapped in MS patients at an early disease stage and compared between disease phenotypes as they evolve with disease progression.

### 4.6 Conclusion

Widespread WM and GM atrophy is present in recently-diagnosed RRMS, suggesting neurodegeneration across tissue types. This is observed particularly in the brainstem, cerebellar GM, subcortical regions and temporal-occipital GM and NAWM, which accords with limited

previously published data in other MS cohorts. Analyses based on volumetry and VBM demonstrate different patterns of atrophy, albeit with some regional overlap. Atrophy measures targeted to these specific brain regions may provide improved markers of neurodegeneration, and potential future imaging stratifiers of disease progression and endpoints for therapeutic trials. Our future aims are directed at mapping neurodegenerative patterns across a ten-year time-period in the FutureMS cohort, as well as correlating regional atrophy with evolving clinical disability and other imaging and liquid biomarkers of neurodegeneration available in FutureMS.

## Supporting information

**S1 Table. Future MS MRI parameters for protocol A and B.**
(DOCX)

**S2 Table. Statistical parameters for linear mixed-effects models evaluating the effect of DMT use at w1 (yes/no) on brain tissue volume change over time.**
(DOCX)

**S3 Table. Detailed statistical parameters for linear mixed-effect models evaluating the effect of time on brain volumes.**
(DOCX)

**S4 Table. Regional GM and NAWM volume results for change over time (w1-w0).**
(DOCX)

## Acknowledgments

We would like to thank non-author contributors of the FutureMS consortium as follows: Amit Akula, Sergio Baranzini, Fiona Barret, Mark Bastin, Chris Batchelor, Emily Beswick, Fraser Brown, Tracy Brunton, Javier Carod Artal, Jessie Chang, Yingdi Chen, Shuna Colville, Annette Cooper, Denise Cranley, Rachel Dakin, Baljean Dhillon, Elizabeth Elliott, James Finlayson, Peter Foley, Stella Glasmacher, Angus Grossart, Haane Haagenrud, Katarzyna Hafezi, Emily Harrison, Adil Harroud, Sara Hathorn, Tracey Hopkins, Aidan Hutchison, Charlotte Jardine, Kiran Jayprakash, Matt Justin, Patrick Kearns, Gwen Kennedy, Lucy Kessler, Michaela Kleynhans, Juan Larraz, Katherine Love, Dawn Lyle, James MacDonald, Jen MacFarlane, Lesley Macfarlane, Alan Maclean, Bev MacLennan, Margaret-Ann MacLeod, Nicola Macleod, Don Mahad, Sarah-Jane Martin, Conni McCarthy, Ian Megson, Daisy Mollison, Mary Monaghan, Lee Murphy, Katy Murray, Judith Newton, Julian Ng Kee Kwong, Jonathan O'Riordan, David Perry, Suzanne Quigley, Adam Scotson, Scott Semple, Amy Stenson, Christine Weaver, Stuart Webb, Belinda Weller, Anna Williams, Stewart Wiseman, Charis Wong, Michael Wong and Rosie Woodward. With special thanks to all FutureMS participants who have made this study possible. For the purpose of open access, the author has applied a Creative Commons Attribution (CC BY) licence to any Author Accepted Manuscript version arising from this submission.

## Author Contributions

**Conceptualization:** Rozanna Meijboom, Adam D. Waldman.

**Data curation:** Rozanna Meijboom, Agniete Kampaite, Nicole White.

**Formal analysis:** Rozanna Meijboom.

**Funding acquisition:** Peter Connick, Siddharthan Chandran.

**Methodology:** Rozanna Meijboom, Michael J. Thrippleton.

**Project administration:** N. J. J. MacDougall, Peter Connick, Siddharthan Chandran.

**Resources:** David P. J. Hunt, Adam D. Waldman.

**Software:** Elizabeth N. York, Mathew A. Harris, Maria del C. Valdés Hernández.

**Supervision:** Adam D. Waldman.

**Visualization:** Rozanna Meijboom.

**Writing – original draft:** Rozanna Meijboom.

**Writing – review & editing:** Rozanna Meijboom, Elizabeth N. York, Agniete Kampaite, Mathew A. Harris, Maria del C. Valdés Hernández, Michael J. Thrippleton, N. J. J. Mac-Dougall, David P. J. Hunt, Adam D. Waldman.

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
