## [Decision Letter · Decision Letter 0]

26 Jan 2023

PONE-D-22-30246Patterns of brain degeneration in early-stage relapsing-remitting multiple sclerosisPLOS ONE

Dear Dr. Meijboom,

Thank you for submitting your manuscript to PLOS ONE. After careful consideration, we feel that it has merit but does not fully meet PLOS ONE’s publication criteria as it currently stands. Therefore, we invite you to submit a revised version of the manuscript that addresses the points raised during the review process. As you will see, both Reviewers have major concerns regarding several methodological aspects of your study. There is a substantial amount of work to address these issues but all appear to be doable.

We look forward to receiving your revised manuscript.

Kind regards,

Niels Bergsland

Academic Editor

PLOS ONE

Journal Requirements:

"Additional funding for authors came from the MS Society Edinburgh Centre for MS Research (grant reference 133; RM, AK), Chief Scientist Office – SPRINT MND/MS program (ENY), Anne Rowling Regenerative Neurology Clinic (ENY), NHS Lothian Research and Development Office (MJT), the Row Fogo Charitable Trust (grant reference BROD.FID3668413; MVH), and the UK Dementia Research Institute (SC), which receives its funding from UK DRI Ltd, funded by the UK Medical Research Council, Alzheimer’s Society and Alzheimer’s Research UK. Additional funding for the Edinburgh university 3T MRI Research scanner in RIE is funded by the Wellcome Trust (104916/Z/14/Z), Dunhill Trust (R380R/1114), Edinburgh and Lothians Health Foundation (2012/17), Muir Maxwell Research Fund, Edinburgh Imaging, and University of Edinburgh."

 "With thanks to FutureMS, hosted by Precision Medicine Scotland Innovation Centre (PMS-IC) and funded by a grant from the Chief Scientist Office, Scotland, to PMS-IC and Biogen Idec Ltd Insurance (combined funding under reference Exemplar SMS_IC010). Additional funding for authors came from the MS Society Edinburgh Centre for MS Research (grant reference 133; RM, AK), Chief Scientist Office – SPRINT MND/MS program (ENY), Anne Rowling Regenerative Neurology Clinic (ENY), NHS Lothian Research and Development Office (MJT), the Row Fogo Charitable Trust (grant reference BROD.FID3668413; MVH), and the UK Dementia Research Institute (SC), which receives its funding from UK DRI Ltd, funded by the UK Medical Research Council, Alzheimer’s Society and Alzheimer’s Research UK. Additional funding for the Edinburgh university 3T MRI Research scanner in RIE is funded by the Wellcome Trust (104916/Z/14/Z), Dunhill Trust (R380R/1114), Edinburgh and Lothians Health Foundation (2012/17), Muir Maxwell Research Fund, Edinburgh Imaging, and University of Edinburgh. The funders had no role in study design, data collection and analysis, decision to publish, or preparation of the manuscript."

3.We note that you have indicated that data from this study are available upon request. PLOS only allows data to be available upon request if there are legal or ethical restrictions on sharing data publicly. For more information on unacceptable data access restrictions, please see http://journals.plos.org/plosone/s/data-availability#loc-unacceptable-data-access-restrictions.

Reviewers' comments:

Reviewer's Responses to Questions

**Comments to the Author**

1. Is the manuscript technically sound, and do the data support the conclusions?

Reviewer #1: Partly

Reviewer #2: Partly

2. Has the statistical analysis been performed appropriately and rigorously? 

Reviewer #1: Yes

Reviewer #2: Yes

3. Have the authors made all data underlying the findings in their manuscript fully available?

Reviewer #1: No

Reviewer #2: No

4. Is the manuscript presented in an intelligible fashion and written in standard English?

Reviewer #1: Yes

Reviewer #2: Yes

5. Review Comments to the Author

Reviewer #1: Meijboom et al. performed an MRI evaluation of short-term atrophy progression over 1 year of follow-up in newly diagnosed relapsing-remitting MS patients using established segmentation methods. They showed widespread GM and NAWM atrophy occurring in the brainstem, cerebellar GM, and subcortical and occipital-temporal regions, which is in line with previous studies performed in this field. However, I do have several concerns mainly regarding the methodological aspects of this study.

- Page 4, line 62-63: “There is currently no cure for MS and treatments targeting the neurodegenerative aspects of MS are limited.”; page 4, line 72-73 “However, these appear to have only a limited effect on reducing associated neurodegenerative processes”. These two sentences contain - more or less - the same statement. The authors should consider rephrasing.

- Page 4, line 100: “This allows for identification of brain areas specifically affected by neurodegeneration”. This is not entirely true in my point of view. In view of the population studied and the time points (directly after MS diagnosis) selected by the investigators, changes in regional CNS volumes can be influenced by several factors, including inflammation and DMTs (see also next comments). Authors should consider rephrasing.

- In general, the introduction could better highlight the points, that have not been addressed so far by previous studies, as well as how this study aims to fill these knowledge gaps. The aims of this study with regard to these knowledge gaps should be also stated more concretely.

- Page 5, line 107. “Exclusion criteria were intake of DMTs prescribed prior to baseline assessment”. Please explain the rationale for this decision.

- In my point of view the selected time points in this study pose a major methodological issue for this work. The investigators included only treatment naive patients at baseline. Then, during the observation time the majority of patients received DMT for the first time (as shown in table 2). DMTs have a significant effect on CNS volumetric measurements, especially in patients with significant ongoing inflammation as the patient population included in this work (recruited directly after a relapse leading to the diagnosis), as demonstrated in previous clinical trials in MS; as these immunomodulatory agents begin to reduce inflammatory activity in the CNS, they lead to an accelerated reduction of brain volumes, which is not thought to be related to neurodegeneration, also known as pseudoatrophy. Thus, the observed temporal differences between time points might well be due to a process not related to neurodegeneration, as the investigators claim, which is also the primary aim of this study. In addition, the authors do not mention how many patients started each DMT agent. High efficacy DMT could have produced a larger pseudoatrophy effect. Even though the investigators show no differences in volume changes over time between treated and untreated patients, selection bias is introduced in this analysis, since patients with a more severe clinical course are most likely to be treated and to receive higher-efficacy DMT.

- As the authors also state in the study limitations, the absence of a control group to compare temporal volumetric changes is an important limitation in this study. Volumetric changes do take place due to “healthy” aging, are accelerated in older subjects and cannot be easily disentangled from MS-related neurodegeneration without a control group, especially in the way the investigators conducted their analysis (i.e. longitudinal voxel-based morphometry). Did the authors enter the interaction term between baseline age and follow-up time in their statistical models in order to at least account for the aging effect during the observation period? If not, it could be that a part of the results regarding volume reduction across different CNS studies is to be ascribed to aging and not ongoing MS-pathology.

- Page 10, line 196-197: “WML volume change was added as a covariate to correct for GM and WM volume changes induced by expanding neuroinflammation”. WML can also shrink (e.g. as a result of remyelination). In addition, changes in WML volumes can also be influenced by factors other than neuroinflammation. The authors should consider rephrasing.

- Table 2, Supplementary Tables 2-4: the regression coefficients are displayed in absolute volumes, which are hard to interpret. The authors should consider a presentation of their results in way that allows for better understanding from the reader (e.g. percentages).

- The authors are sometimes vague regarding their results. For instance: Page 13, lines 252-253: “Site and WML change were significant for some regions…”. These results should be also at least briefly summed up. The authors cannot expect the reader to go through several pages of tables to identify the study results.

- What was the rational for conducting both voxel-based morphometry and FreeSurfer segmentation and corresponding analysis. The authors also state (page 16, line 298-299): “In addition, the effects of different imaging analysis approaches were compared.” and (page 16, line 311-313): However, overall, fewer regions of GM change were observed than with the volumetric approach, particularly in the parietal and frontal lobes, without prominent hemispheric differences.” How do the authors interpret this?

- The authors state (page 3, line 52-54): “Atrophy measures targeted to these most severely affected regions may be more sensitive and specific than whole-brain atrophy as imaging stratifiers and endpoints for clinical decision making and therapeutic trials.” However, the analysis of imaging data conducted in this study does not allow for identification of the most severely affected brain regions, but rather highlight, which regions are affected (even minimally). In order to pinpoint “hot spots” of brain atrophy, a comparison of volumetric changes over time between brain regions should have been conducted. I believe this study could profit from the addition of such an analysis.

Reviewer #2: The current work aims to investigate regional patterns of neurodegeneration in RRMS and compare standard volumetry with a voxel-based morphometry approach. The presented work is interesting, but not very novel as regional patterns of neurodegeneration have been investigated before. One major concern about this work is that a lot of atrophy was found within only 1 year. In this early phase of MS neurodegeneration is expected to be mild, especially within this short time period. Methodologically, the results could have been influenced by (the resolution of) neuroinflammation. In fact, 2/3 of the cohort started DMT during the assessed year, so the large ‘atrophy’ found could have been pseudo-atrophy. Finally, although the concept is certainly of interest, the pipeline is rather simple and requires major improvements, including the use of lesion filling and using the same segmentation method for the volumetric and VBM approach.

Abstract

• Provide some information on the demographics of the cohort, how “early-stage” is the RRMS population?

Introduction

• In general the introduction is not very well structured, for example the effect of DMT on neurodegenerative processes is mentioned in both first and second part, try to integrate this.

• It is not very relevant to note in the first part that MS prevalence is particularly high in Scotland, this does not lead up to this particular study (which is on disease severity, not prevalence, and does not compare findings with other regions).

• “Less regional detail appears to be known about macrostructural WM volume”, rephrase? This is vague.

• It is not introduced why it would be relevant to look at regional NAWM loss.

Methods

• The intracranial volume (ICV) estimation based on manually edited brain masks is not a standard way to correct for head size. This should be replaced, for example by the ICV estimated by FreeSurfer (eTIV).

• “The edited w0 intracranial image was registered to the w1 T1W image”. So a skull-stripped image was registered to an image with skull? This is not correct, images should both be skull-stripped.

• The input for FreeSurfer should be lesion filled T1w images, subtracting WML masks from the tissue segmentations is not a correct method to do this, as this will not influence the original T1 images and the segmentation itself is already influenced by these lesions. Tissue segmentations are effected by presence of WML. Use a lesion filling approach on the T1w images and repeat the FreeSurfer analysis with the filled T1w images.

• Not a logical choice to calculate the tissue masks using fslstats, partial volume effects are not taking into account this way. Freesurfer provides volume measures as output (stats/aseg.stats), so please use this for the tissue volumes.

• Since WML masks were substracted from the tissue segmentations, the decrease in regional NAWM could be confounded by increasing lesion volumes. Repeat the analysis with total WM volumes instead of only NAWM.

• The chosen WML segmentation method is not standardly used in the field. As such it is important to provide some more details why this was used with references of a validation study or use a more widely used method for this.

• For the VBM analysis a different segmentation method was used compared with the volumetric analysis with FreeSurfer. In order to compare these approaches, the same segmentation method should be used, this could also explain the discrepancy in results. In addition, input for VBM pipeline should also be the lesion-filled T1w images.

• Why was WML change included in the model and not the WML at w0 and w1? Now the WML change is considered a constant factor over time which is not the case.

• “For each outcome variable, extreme outliers (>3 SD) as well as participants with missing data at either time point were excluded”. Since visual inspection was performed, it should not be necessary to remove extreme outliers from the data, since these then are not errors but actual findings. In addition, subjects with missing timepoints do not have to be removed, as mixed effects analysis can also deal with missing datapoints.

• The abbreviation FDR is not introduced.

• “A general linear model (GLM) design matrix was created with GM difference, age, sex, imaging site, DMT status at w1 and subject-specific WML mask as explanatory variables (EV).” “WMLs were included as EVs to ensure their exclusion from the GM images.” This is not an appropriate method to exclude WML from GM images, lesion filling should have been applied (see earlier comments).

• Imputing missing DMT status with “yes” is not the preferred approach to deal with missing values in the mixed effects analysis, would be better to insert it as a missing value in the mixed effects analysis as the number of subjects with missing data is small.

• “Results were reported for voxels at p<0.001 and family-wise error (FWR) corrected for multiple comparisons.” Which p-value was chosen as significant threshold for FWR?

Results

• Findings are interesting but as mentioned can be significantly affected by the lack of lesion filling and the induction of DMT, as well as methodological differences between the two pipelines. This impairs interpretation of the data currently. Moreover, healthy controls are not available so it’s not possible to compare found atrophy rates of MS with healthy controls.

• No information given in methods how the FreeSurfer segmentation quality control was exactly performed. Also, would be better to harmonize the subjects included in the FreeSurfer analyses with the VBM analyses since you are aiming for a direct comparison of the methods.

• Table 2: Provide numbers how many patients were on which DMT. also provide baseline whole brain and GM volumes. Was the WML volume between w0 and w1 significantly different? Please statistically test this.

• As mentioned, a large number of patients started DMT between w0 and w1, especially for high efficacy DMT’s it’s known that this can cause pseudo-atrophy in the first year. Please take this into account in the analysis or describe how you deal with this confounding effect.

• Table 3: not clear if presented p-values were corrected or not, if not, please provide corrected p-values (q-values).

• Since the aim of the paper is to compare the volumetric and VBM approach, it would be helpful to provide a table/figure comparing the regions found by both approaches. It will be rather hard for the reader now to see the overlap between the two approaches since figure 3 does not include subcortical areas.

Discussion

• Discussion is a bit generic but overall ok. The discussion could be improved by adding which atrophy rates you would expect for the different brain regions based on earlier studies and if the found atrophy rates are in line with that. Especially in terms of the relatively short follow-up of 1 year and early stage of the disease, reflect if your results are in line with literature.

• What is cortical NAWM?

• It is important to discuss in more detail why you think the volumetric approach found more atrophic regions compared with the VBM approach. This is not expected, how do you interpret and justify this difference?

• “Surprisingly, we also did not detect volumetric changes within areas in the frontal lobe associated with sensorimotor functioning.” Explain in more detail why you expected this, is the other early MS cohort that found this comparable to yours in terms of disease duration and disability?

• Do you have any recommendations which method should be used/is best to assess regional atrophy? What are the main advantages/disadvantages of both methods?

Conclusion

• “Atrophy measures targeted to the most affected regions may provide more sensitive and specific markers of neurodegeneration, as imaging stratifiers of disease progression and endpoints for future therapeutic trials.” Regional measures being more sensitive biomarkers than global ones has not been discussed in the discussion and has not been tested in this study.

• “Analyses based on volumetry and VBM demonstrate different patterns of atrophy, albeit with some regional overlap.” Clarify which regions were most affected according to both analyses and what the differences where.

6. PLOS authors have the option to publish the peer review history of their article (what does this mean?). If published, this will include your full peer review and any attached files.

Reviewer #1: **Yes: **Charidimos Tsagkas

Reviewer #2: No

---

## [Author Response · Author response to Decision Letter 0]

17 May 2023

We thank the reviewers for their helpful comments and queries, which we have addressed on a point by point basis below, and referenced changes to manuscript in response to these.

Review Comments to the Author

Reviewer #1 (Charidimos Tsagkas): 

Meijboom et al. performed an MRI evaluation of short-term atrophy progression over 1 year of follow-up in newly diagnosed relapsing-remitting MS patients using established segmentation methods. They showed widespread GM and NAWM atrophy occurring in the brainstem, cerebellar GM, and subcortical and occipital-temporal regions, which is in line with previous studies performed in this field. However, I do have several concerns mainly regarding the methodological aspects of this study.

1. Page 4, line 62-63: “There is currently no cure for MS and treatments targeting the neurodegenerative aspects of MS are limited.”; page 4, line 72-73 “However, these appear to have only a limited effect on reducing associated neurodegenerative processes”. These two sentences contain - more or less - the same statement. The authors should consider rephrasing.

We agree with the reviewer and have now removed the second part of “There is currently no cure for MS and treatments targeting the neurodegenerative aspects of MS are limited.” (line 62-63).

2. Page 4, line 100: “This allows for identification of brain areas specifically affected by neurodegeneration”. This is not entirely true in my point of view. In view of the population studied and the time points (directly after MS diagnosis) selected by the investigators, changes in regional CNS volumes can be influenced by several factors, including inflammation and DMTs (see also next comments). Authors should consider rephrasing.

We agree that atrophy is a downstream and potentially confounded marker of neurodegeneration. It is, however, a pragmatic and recommend measure by MAGNIMSa, which is why we chose to study it here. We have modified the sentence addressed by the reviewer’s comment and moved the focus to atrophy rather than neurodegeneration. Additionally, we have made further changes throughout the introduction, and adjusted the title, moving the focus to atrophy, which should more accurately reflect the findings of this study.

“This allows for identification of brain areas affected by atrophy, which may provide possible biomarkers of disease progression for DMT trials.”

Reference:

aSastre-Garriga J, Pareto D, Battaglini M, Rocca MA, Ciccarelli O, Enzinger C, et al. MAGNIMS consensus recommendations on the use of brain and spinal cord atrophy measures in clinical practice. Nat Rev Neurol [Internet]. 2020 Mar 24;16(3):171–82. Available from: http://www.nature.com/articles/s41582-020-0314-x

3. In general, the introduction could better highlight the points, that have not been addressed so far by previous studies, as well as how this study aims to fill these knowledge gaps. The aims of this study with regard to these knowledge gaps should be also stated more concretely.

We have rephrased the introduction to better highlight the gaps in the literature and state the aims of this study. We hope that this is now sufficiently clearer.

“Current literature does not provide detailed insights into regional WM volume loss in recently-diagnosed RRMS, and only a limited number of studies have examined specific regional GM volume in early stages of the disease (11,16–18). The aim of this study was to investigate neurodegenerative changes reflected in loco-regional atrophy in recently-diagnosed people with RRMS. Additionally, two analysis approaches were used to understand disparate results in the literature and the degree to which they result from different measurement methodology. Baseline and 1-year follow-up MR imaging data was used from the FutureMS study (40,41), a large multicentre cohort of RRMS patients within six months after diagnosis. This allows for identification of brain areas affected by early atrophy in RRMS, which may provide possible biomarkers of disease progression for DMT trials.”

4. Page 5, line 107. “Exclusion criteria were intake of DMTs prescribed prior to baseline assessment”. Please explain the rationale for this decision.

We have now added the rationale for this decision under section 2.1. 

“Exclusion criteria were participation in a clinical trial prior to baseline assessment, contraindications for MRI, and intake of DMTs prescribed prior to baseline assessment. The latter criteria was selected because DMT intake in the Scottish population is very low compared with other countries, creating a unique opportunity to develop natural history predictive tools.”

5. In my point of view the selected time points in this study pose a major methodological issue for this work. The investigators included only treatment naive patients at baseline. Then, during the observation time the majority of patients received DMT for the first time (as shown in table 2). 

DMTs have a significant effect on CNS volumetric measurements, especially in patients with significant ongoing inflammation as the patient population included in this work (recruited directly after a relapse leading to the diagnosis), as demonstrated in previous clinical trials in MS; as these immunomodulatory agents begin to reduce inflammatory activity in the CNS, they lead to an accelerated reduction of brain volumes, which is not thought to be related to neurodegeneration, also known as pseudoatrophy. Thus, the observed temporal differences between time points might well be due to a process not related to neurodegeneration, as the investigators claim, which is also the primary aim of this study. 

In addition, the authors do not mention how many patients started each DMT agent. High efficacy DMT could have produced a larger pseudoatrophy effect. Even though the investigators show no differences in volume changes over time between treated and untreated patients, selection bias is introduced in this analysis, since patients with a more severe clinical course are most likely to be treated and to receive higher-efficacy DMT.

We respectfully disagree with the reviewer that the design of this study poses major methodological issues. The issues the reviewer describes, such as selected time points, treatment naïve patients at baseline and the majority receiving DMTs for the first time before 1-year follow-up, are innate features of this study. FutureMS is a clinical study, investigating patients with RRMS just after diagnosis and observing changes within the first year after diagnosis. As we follow a clinical cohort here, it is unsurprising patients receive treatment within the first year after RRMS diagnosis. In order to minimise treatment induced variability within the cohort, we aimed to recruit only those patients who were treatment-naïve for baseline visits. This creates an equal starting point for all participants. Overall, any changes observed in this study using data from FutureMS, will be reflective of a real population of RRMS patients in a very early stage of their disease. 

We agree with the reviewer that use of DMT may be a confounder and can lead to pseudoatrophy. In order to minimise this effect, all statistical analyses performed included DMT status at follow-up as a covariate. DMT as covariate was not significant for any of the global areas as well as cerebellum, brainstem or subcortical areas. For the WM and GM regions, DMT was only significant for 6 out of 105 regions that showed significant atrophy. Furthermore, we performed an analysis specifically looking at differences in WB volume change over time between pwRRMS on DMT and those not on DMT. Effect of DMT status was not found (results included in the supplement). Overall, these results suggest that there is no bias due to pseudoatrophy and that in case of an effect – 6/105 regions – the model has appropriately adjusted for this. It is not, however, possible to entirely exclude the pseudoatrophy-induced bias not caused by DMTs, i.e. spontaneous resolution of inflammation and hence swelling. We have added this and the rationale for DMTs to the limitation section in the discussion. 

“Third, pseudoatrophy may have been caused by spontaneous recovery from inflammation and hence swelling. This cannot be corrected for and should be taken into consideration when interpreting results. Pseudoatrophy may also have been caused by DMTs, which we have carefully corrected for by including adding DMT as a covariate. Results showed that DMTs did not have a significant effect on whole-brain, cerebral, cerebellar, brainstem or subcortical atrophy, and only in 6 out of 105 WM/GM regions that showed significant atrophy, which was corrected for accordingly.”

We agree with the reviewer that it would be helpful for the reader to be aware how many patients started each DMT agent and have added this to the manuscript.

“Participants were DMT-naïve at w0, but most started receiving DMTs between w0 and w1 (Table 2). Post-hoc regression analyses showed tissue volume change was not significantly different between participants positive or negative for DMTs at w1 (supplement Table 2). The majority of DMTs administered to this cohort consisted of dimethyl fumarate (97 volumetrics sample; 104 VBM sample), but glatirimer acetate (31; 37), beta interferon (16; 20) and alemtuzumab (22; 24) were also used. A few participants received natalizumab (8 volumetrics sample; 9 VBM sample), fingolimod (7; 7), teriflunomide (5; 9), azathioprine (1; 1), a combination of the aforementioned DMTs (16; 18) or a DMT not further specified (21; 24).”

6. As the authors also state in the study limitations, the absence of a control group to compare temporal volumetric changes is an important limitation in this study. Volumetric changes do take place due to “healthy” aging, are accelerated in older subjects and cannot be easily disentangled from MS-related neurodegeneration without a control group, especially in the way the investigators conducted their analysis (i.e. longitudinal voxel-based morphometry). Did the authors enter the interaction term between baseline age and follow-up time in their statistical models in order to at least account for the aging effect during the observation period? If not, it could be that a part of the results regarding volume reduction across different CNS studies is to be ascribed to aging and not ongoing MS-pathology.

We agree with the reviewer that we have not corrected for an ageing effect, i.e. older people having more atrophy than younger people. We have now rerun all the volumetrics models with the interaction age*time point added in and observed that after multiple comparison correction, this interaction was significant in three regions: left cerebellar GM, and left and right superior frontal NAWM. In these three regions volume change over time is dependent on age. The interaction effect was excluded from models where it showed not significant. We have made changes in the text, tables and supplement tables accordingly. The limitation of not having control data available for this study has been previously described in the limitations section.

Section 2.4.1

“Additionally, to correct for any differences in volume change over time due to older and younger participants, the interaction between time and age was added. The interaction was removed from the model if not significant.”

Section 2.4.2

“The interaction of age and time was not added here. Given the interaction was not significant in 98% of volumetric regions studied, it is unlikely it will have a significant effect on the overall VBM results.”

Section 3.2

“Additionally, atrophy rates for the left cerebellar GM were significantly different across age.”

Section 3.3

“Additionally, 2/14 frontal NAWM regions, i.e. the left and right superior frontal NAWM, showed differential atrophy rates across age.”

7. Page 10, line 196-197: “WML volume change was added as a covariate to correct for GM and WM volume changes induced by expanding neuroinflammation”. WML can also shrink (e.g. as a result of remyelination). In addition, changes in WML volumes can also be influenced by factors other than neuroinflammation. The authors should consider rephrasing.

We agree with the reviewer and have rephrased this sentence accordingly.

“WML volume change was added as a covariate to correct for GM and WM volume changes induced by enlarging and/or receding WMLs.”

8. Table 2, Supplementary Tables 2-4: the regression coefficients are displayed in absolute volumes, which are hard to interpret. The authors should consider a presentation of their results in way that allows for better understanding from the reader (e.g. percentages).

The regression coefficients (B values) shown in Supplement Tables 2-4 and main text Table 3 are standardised regression coefficients, and hence not displayed in absolute volumes. We agree with the reviewer, however, that this was not clear and have made changes to the tables accordingly to explain and emphasise that the shown regression coefficients are standardised. 

Additionally, we would like to note that in Table 3 and Supplementary Table 4, percentage change over time was already shown for each brain region assessed. This was done to allow the reader easier interpretation of the observed volume changes over time.

9. The authors are sometimes vague regarding their results. For instance: Page 13, lines 252-253: “Site and WML change were significant for some regions…”. These results should be also at least briefly summed up. The authors cannot expect the reader to go through several pages of tables to identify the study results.

We agree with the reviewer that the covariate results were not sufficiently clear in the text. The reason for doing this, however, was to avoid too much text on additional results, which distracts from focus on the main results. We ensured all results, including those for covariates, were included in tables to provide transparency. To meet the reviewer’s request, however, we have added detail on covariates to the section on p.13 as described in their comment. 

“Covariate significance (p<0.05) was variable across regions. Age and sex were significant in nearly all regions, except the bilateral cerebral NAWM (age and sex), brainstem (sex), right cerebellar GM (sex), and bilateral cerebellar NAWM (age). Site was significant for the bilateral amygdala, left accumbens, right cerebellar GM, bilateral cerebral GM, bilateral hippocampus, bilateral cerebellar NAWM, left cerebral NAWM, left thalamus (site), and whole-brain (site). WML change was significant for the bilateral amygdala and right thalamus only. DMT status at w1 did not reach significance for any brain region.”

10. What was the rational for conducting both voxel-based morphometry and FreeSurfer segmentation and corresponding analysis. The authors also state (page 16, line 298-299): “In addition, the effects of different imaging analysis approaches were compared.” and (page 16, line 311-313): However, overall, fewer regions of GM change were observed than with the volumetric approach, particularly in the parietal and frontal lobes, without prominent hemispheric differences.” How do the authors interpret this?

The rationale for comparing VBM and FreeSurfer was to understand disparate results in the literature and the degree to which they result from different measurement methodology. We agree with the reviewer and realise we have not sufficiently explained this in the manuscript. We have now added further clarification to the introduction. 

“The aim of this study was to investigate neurodegenerative changes reflected in loco-regional atrophy in recently-diagnosed people with RRMS. Additionally, two analysis approaches were used to understand disparate results in the literature and the degree to which they result from different measurement methodology. “ 

The interpretation of the observed differential results between VBM and FreeSurfer were described in the discussion of the manuscript. This can be found under section ‘4.4 Volumetry and VBM’. 

11. The authors state (page 3, line 52-54): “Atrophy measures targeted to these most severely affected regions may be more sensitive and specific than whole-brain atrophy as imaging stratifiers and endpoints for clinical decision making and therapeutic trials.” However, the analysis of imaging data conducted in this study does not allow for identification of the most severely affected brain regions, but rather highlight, which regions are affected (even minimally). In order to pinpoint “hot spots” of brain atrophy, a comparison of volumetric changes over time between brain regions should have been conducted. I believe this study could profit from the addition of such an analysis.

We agree with the reviewer it would be interesting to identify hot spots of atrophy by comparing volumetric changes over time between brain regions. This was not the aim of this study, however, and we apologise for having been unclear about this. We have rephrased the above text, as well as the corresponding sentence in the discussion, to accurately reflect the results of the study. Additionally, we would like to refer to Table 3 and Supplement Table 4, which include mean percentage volume change for each explored brain region. Although not a formal analysis of atrophic hot spots, they can give the reader an idea of which regions appear most affected.

“Atrophy measures targeted to specific brain regions may provide improved markers of neurodegeneration, and potential future imaging stratifiers and endpoints for clinical decision making and therapeutic trials..”

 

Reviewer #2: 

The current work aims to investigate regional patterns of neurodegeneration in RRMS and compare standard volumetry with a voxel-based morphometry approach. The presented work is interesting, but not very novel as regional patterns of neurodegeneration have been investigated before. One major concern about this work is that a lot of atrophy was found within only 1 year. In this early phase of MS neurodegeneration is expected to be mild, especially within this short time period. Methodologically, the results could have been influenced by (the resolution of) neuroinflammation. In fact, 2/3 of the cohort started DMT during the assessed year, so the large ‘atrophy’ found could have been pseudo-atrophy. Finally, although the concept is certainly of interest, the pipeline is rather simple and requires major improvements, including the use of lesion filling and using the same segmentation method for the volumetric and VBM approach.

Abstract

1. Provide some information on the demographics of the cohort, how “early-stage” is the RRMS population?

We have added some clarification on this to the method sections of the abstract (see below as well). Additionally, we have changed the wording of early-stage throughout the document to recently-diagnosed to further clarify this point. 

“RRMS patients (N = 354) underwent 3T structural MRI <6 months after diagnosis and at 1-year follow-up, as part of the Scottish multicentre ‘FutureMS’ study.”

Introduction

2. In general the introduction is not very well structured, for example the effect of DMT on neurodegenerative processes is mentioned in both first and second part, try to integrate this.

We agree with the reviewer that the topic of DMT and neurodegeneration should not be addressed twice and amended the text accordingly. Please also see our answer to R1.C1. We are unsure, however, what further sections of the introduction are not sufficiently structured and would be happy to address any specific examples that the reviewer might have in mind.

3. It is not very relevant to note in the first part that MS prevalence is particularly high in Scotland, this does not lead up to this particular study (which is on disease severity, not prevalence, and does not compare findings with other regions).

We mentioned this as our study participants are recruited from the Scottish population. We agree with the reviewer, however, that it is not essential to mention the Scottish prevalence of MS in the introduction and have removed the sentence accordingly.

4. “Less regional detail appears to be known about macrostructural WM volume”, rephrase? This is vague.

We have edited this sentence for clarification.

“Less is known about regional patterns of WM atrophy in RRMS, but WM atrophy also occurs in early disease and has been suggested to occur independently of WML development (10,19,20).”

5. It is not introduced why it would be relevant to look at regional NAWM loss.

We agree with the reviewer we should have been clearer about this and added a sentence to the introduction to address why it is relevant to study neurodegeneration in NAWM, in terms of atrophy.

“Less is known about regional patterns of WM atrophy in RRMS, but WM atrophy also occurs in early disease and has been suggested to occur independently of WML development (10,19,20). Furthermore, previous studies have shown microstructural changes reflective of neurodegeneration within the normal-appearing WM (NAWM) (21).” 

Methods

6. The intracranial volume (ICV) estimation based on manually edited brain masks is not a standard way to correct for head size. This should be replaced, for example by the ICV estimated by FreeSurfer (eTIV).

We have estimated intracranial volume (ICV) based on intracranial masks extracted with widely-used FSL BET2. All these masks were manually checked and edited where non-intracranial tissue had been included by the software, hence improving the final masks on which the ICV is based. We do not see any objections to using this method and respectfully disagree with the reviewer that a switch to FreeSurfer intracranial masks should be made.

7. “The edited w0 intracranial image was registered to the w1 T1W image”. So a skull-stripped image was registered to an image with skull? This is not correct, images should both be skull-stripped.

Intracranial image at w0 was indeed registered to the w1 T1W image including the skull. All registrations were checked and approved. Could we ask the reviewer to clarify why they suggest this is incorrect?

8. The input for FreeSurfer should be lesion filled T1w images, subtracting WML masks from the tissue segmentations is not a correct method to do this, as this will not influence the original T1 images and the segmentation itself is already influenced by these lesions. Tissue segmentations are effected by presence of WML. Use a lesion filling approach on the T1w images and repeat the FreeSurfer analysis with the filled T1w images.

We of course agree with the reviewer that lesions may affect tissue segmentation. We therefore checked all segmentations manually. For the majority of cases, segmentation was performed accurately. For those cases where segmentations did not work adequately, mask edits were performed in Freeview. Cases were excluded if repeated editing of the output and FreeSurfer reruns remained unsuccessful. We have accounted for the effects of lesions, and therefore do not believe that lesion-filling and a repeat of FreeSurfer processing is required at this stage.

9. Not a logical choice to calculate the tissue masks using fslstats, partial volume effects are not taking into account this way. Freesurfer provides volume measures as output (stats/aseg.stats), so please use this for the tissue volumes.

Unfortunately, we are unsure what the reviewer means by suggesting partial volume effects are not taken into account using fslstats, but are taken into account using FreeSurfer volume output. Our volume extraction is performed on the FreeSurfer output masks, just like the FreeSurfer volume extraction.

10. Since WML masks were subtracted from the tissue segmentations, the decrease in regional NAWM could be confounded by increasing lesion volumes. Repeat the analysis with total WM volumes instead of only NAWM.

We agree with the reviewer that increasing lesion volume may affect atrophy rates and have therefore already corrected for this by including WML volume change as covariate in the volumetric regression models and WML masks as explanatory variable in the VBM model. 

11. The chosen WML segmentation method is not standardly used in the field. As such it is important to provide some more details why this was used with references of a validation study or use a more widely used method for this.

We respectfully disagree with the reviewer that the FLAIR thresholding method is an uncommon method for identifying WMLs; segmentation using thresholding of images is an old concept and there are numerous publications using FLAIR thresholding for lesion segmentation. We refer to the references below for a few examples in the literature. Additionally, we already have ensured to reference the FutureMS protocol paper in which the method is carefully explained, as well as referenced the original paper it was based on. Moreover, as with any lesion segmentation method available, FLAIR thresholding methods have drawbacks and are not 100% accurate. To compensate for this and assure quality, all WML masks have been manually checked and edited where necessary. This has also already been explained in the manuscript. We do not think further explanation is warranted.

1) Valdés Hernández, M.d.C., Ferguson, K.J., Chappell, F.M. et al. New multispectral MRI data fusion technique for white matter lesion segmentation: method and comparison with thresholding in FLAIR images. Eur Radiol 20, 1684–1691 (2010). https://doi.org/10.1007/s00330-010-1718-6

2) Cabezas M., Oliver A., Roura E., Freixenet J., Vilanova J.C., et al. Automatic multiple sclerosis lesion detection in brain MRI by FLAIR thresholding. Computer Methods and Programs in Biomedicine, 115, 147-161 (2014). https://doi.org/10.1016/j.cmpb.2014.04.006.

3) Sahoo P.K, Soltani S., Wong A.K.C. A survey of thresholding techniques. Computer Vision, Graphics, and Image Processing. 41, 233-260 (1988). https://doi.org/10.1016/0734-189X(88)90022-9

4) Pious A. E. and Sridevi U. K. A novel segment white matter hyperintensities approach for detecting Alzheimer. Computer Systems Science and Engineering, 44, 2715–2726 (2023). https://doi.org/10.32604/csse.2023.026582

12. For the VBM analysis a different segmentation method was used compared with the volumetric analysis with FreeSurfer. In order to compare these approaches, the same segmentation method should be used, this could also explain the discrepancy in results. In addition, input for VBM pipeline should also be the lesion-filled T1w images.

We respectfully disagree with the reviewer that the same segmentation method should have been used. The purpose of using two methods was to understand how analysis methodology would emphasise different features of volume change and explain disparate findings between different published studies. We have edited the introduction to reflect this aim more clearly. Changing how one method works, wouldn’t have allowed for a fair comparison. We agree that different analysis methodology can lead to different results from the same data, which is one of our key findings that we discuss in the paper.

13. Why was WML change included in the model and not the WML at w0 and w1? Now the WML change is considered a constant factor over time which is not the case.

We included WML volume change as covariate in the model to correct for volume changes due to WMLs rather than atrophy. We are not sure why adding in WML volume at w0 and w1 as a covariate would be better in a longitudinal model or what the reviewer means by WML change acting is a constant factor, and would like to ask the reviewer for further clarification. 

14. “For each outcome variable, extreme outliers (>3 SD) as well as participants with missing data at either time point were excluded”. Since visual inspection was performed, it should not be necessary to remove extreme outliers from the data, since these then are not errors but actual findings. In addition, subjects with missing timepoints do not have to be removed, as mixed effects analysis can also deal with missing datapoints.

We acknowledge that extreme outliers in our data would have been real findings and not errors. In order to be able to apply the linear model, however, a normal distribution or an approximation thereof is expected. It is therefore that we removed any extreme outliers if present. As far as we are aware, this is a fairly standard approach. 

There are multiple ways of dealing with missing data, each with their own advantages and disadvantages. The mixed effects model in R uses maximum likelihood where it estimates any missing values of the dependent variable, which would be helpful but equally also means creating data that is not there. Another way of dealing with missing data would be imputation. We have chosen, however, to exclude anyone without two time points. As we cannot measure their volume change over time, and loss-to-follow-up was generally due to participants having moved away and not due to worse disease, this seemed a reasonable approach for this particular study. 

15. The abbreviation FDR is not introduced.

The reviewer is correct and we have corrected the mistake.

16. “A general linear model (GLM) design matrix was created with GM difference, age, sex, imaging site, DMT status at w1 and subject-specific WML mask as explanatory variables (EV).” “WMLs were included as EVs to ensure their exclusion from the GM images.” This is not an appropriate method to exclude WML from GM images, lesion filling should have been applied (see earlier comments).

In order to prevent tissue misclassification, we applied subject-specific WML masks to the statistical analysis of VBM output. We have previously explained this in the methods (Methods section 2.4.2). This procedure has been suggested by FSL to deal with lesions/pathologies in the images (https://fsl.fmrib.ox.ac.uk/fsl/fslwiki/Randomise/UserGuide#Missing_Data_and.2For_Lesion_Masking). Moreover, registration during VBM was checked and errors were not detected. We therefore do not believe that lesion-filling is necessary using this alternative approach to classification.

17. Imputing missing DMT status with “yes” is not the preferred approach to deal with missing values in the mixed effects analysis, would be better to insert it as a missing value in the mixed effects analysis as the number of subjects with missing data is small.

As with comment 14 above, there are different ways of dealing with missing data, each with their own advantages and disadvantages. Here, we chose a different approach for missing covariate data than for missing dependent variable data, as covariates are less central to the analysis and we did not want to lose participants who had full dependent variable data available. We chose to replace any missing DMT variables with the mode, one of the many suggested ways of dealing with missing data, as it was a way of keeping these participants in the analysis without changing the data too much. Despite any disadvantages of this method, albeit disadvantages can be equally stated for using maximum likelihood, any effect is expected to be minor, as the number of subjects with missing DMT data is very small and we observed that DMT status did not have a significant effect on the dependent variable.

18. “Results were reported for voxels at p<0.001 and family-wise error (FWR) corrected for multiple comparisons.” Which p-value was chosen as significant threshold for FWR?

We understand our sentence was confusing and have now rephrased it for clarification.

“Results were family-wise error (FWR) corrected for multiple comparisons (p<0.001).”

Results

19. Findings are interesting but as mentioned can be significantly affected by the lack of lesion filling and the induction of DMT, as well as methodological differences between the two pipelines. This impairs interpretation of the data currently. Moreover, healthy controls are not available so it’s not possible to compare found atrophy rates of MS with healthy controls.

As addressed in comments above, we do not believe the lack of lesion-filling has had an effect on our results. Segmentations have all been checked and edited where necessary, and WML volume change (volumetrics) and WML masks (VBM) have been added as covariates to the statistical analyses to correct for WML presence.

We agree that DMT intake may affect atrophy, which is why we ensured to correct for DMT status at follow-up in the statistical models. Furthermore, we performed an analysis specifically looking at differences in WB volume change over time between pwRRMS on DMT and those not on DMT. The results were included in the supplement and showed no effect of DMT status. Although we acknowledge that DMT may be a confounder, we believe we have sufficiently corrected for it here, and dealt with it as much as a clinical study sample allows. We have added a paragraph to the limitations in the discussion to discuss effects of DMTs.

“Third, pseudoatrophy may have been caused by spontaneous recovery from inflammation and associated swelling. This cannot be corrected for, and should be taken into consideration when interpreting results. Pseudoatrophy may also have been caused by DMTs, which we have carefully corrected for by including adding DMT as a covariate. Results showed that DMTs did not have a significant effect on whole-brain, cerebral, cerebellar, brainstem or subcortical atrophy, and only in 6 out of 105 WM/GM regions that showed significant atrophy, which was corrected for accordingly.”

Methodological differences between the two pipelines were exactly the rationale behind comparing them. We realise, however, we have not sufficiently explained this in the introduction and have now added further clarification. Please see below. 

“The aim of this study was to investigate neurodegenerative changes reflected in loco-regional atrophy in recently-diagnosed people with RRMS. Additionally, two analysis approaches were used to understand disparate results in the literature and the degree to which they result from different measurement methodology.” 

We agree with the reviewer that including a healthy control group could provide additional relevant and useful information, and mentioned this previously as a limitation in the discussion of our manuscript. Matched healthy control data were not available for comparison in this study, however, we do not agree that this impairs interpretation of the data. The longitudinal study design mitigates significant variations in brain volume between individuals, and allows regional and global atrophy to be mapped in MS patients at an early disease stage and compared between disease phenotypes as they evolve with disease progression. Additionally, it is worth noting, that overall whole-brain volume decrease observed in this study was 0.5%; this exceeds brain atrophy expected to occur in healthy individuals (0.4%) proposed in the 2020 MAGNIMS guidelines.

20. No information given in methods how the FreeSurfer segmentation quality control was exactly performed. Also, would be better to harmonize the subjects included in the FreeSurfer analyses with the VBM analyses since you are aiming for a direct comparison of the methods.

We agree with the reviewer manual editing of FreeSUrfer output could have been clearer and have now amended the text accordingly.

“Freesurfer output was manually edited by a trained neuroscientist where necessary using FreeView v2.0 (FreeSurfer v6.0) and following FreeSurfer editing guidelines (https://surfer.nmr.mgh.harvard.edu/fswiki/Edits).” 

We are not entirely certain what the reviewer means with harmonising the subjects included in both analyses, however, we suspect they are referring to the VBM analysis having a large sample. We understand this point being made, but are unsure it would be better to reduce the VBM sample size in order to match with the FreeSurfer sample. Both pipelines started with the same sample, but a number of subjects had to be removed from the FreeSurfer analysis due to insufficient quality of segmentations. It seems counterintuitive to then rerun VBM whilst excluding these subjects, despite their data seemingly performing well within the VBM analysis. In any case, it highlights a possible additional difference between the two pipelines, where VBM here seems to cope better with reduced data quality, or alternatively, where FreeSurfer has a better quality check procedure in place. We have added some discussion on this to the manuscript’s discussion under section ‘4.4 Volumetry and VBM’, and hope this has sufficiently addressed the reviewer’s comment.

“Additionally, a larger number of data failed quality checks with FreeSurfer than with VBM, leading to a difference in sample size between the two methods. Although this may suggest that VBM is a more robust method in case of lower data quality, alternatively, this may also indicate that FreeSurfer has a better and stricter quality-check procedure in place.”

21. Table 2: Provide numbers how many patients were on which DMT. also provide baseline whole brain and GM volumes. Was the WML volume between w0 and w1 significantly different? Please statistically test this.

We agree with the reviewer that adding numbers to DMTs would be helpful and have amended the text associated with table 2 accordingly.

“The majority of DMTs administered to this cohort consisted of dimethyl fumarate (97 volumetrics sample; 104 VBM sample), but glatirimer acetate (31; 37), beta interferon (16; 20) and alemtuzumab (22; 24) were also used. A few participants received natalizumab (8 volumetrics sample; 9 VBM sample), fingolimod (7; 7), teriflunomide (5; 9), azathioprine (1; 1), a combination of the aforementioned DMTs (16; 18) or a DMT not further specified (21; 24).”

We also agree with the reviewer that baseline whole-brain and GM volumes are helpful for the reader. These can be found in Table 3. We are not entirely sure what testing the WML volume difference between baseline and follow-up would add to this paper, as the focus of the paper is on atrophy, and have therefore not done this previously. We would be happy to perform the analysis, but would like to ask the reviewer for further clarification for wanting to include this.

22. As mentioned, a large number of patients started DMT between w0 and w1, especially for high efficacy DMT’s it’s known that this can cause pseudo-atrophy in the first year. Please take this into account in the analysis or describe how you deal with this confounding effect.

We absolutely agree with the reviewer that treatment may confound results. In order to adjust for this confounding effect, we used DMT status at follow-up as a covariate in both volumetrics and VBM analyses. In addition, we performed a supplementary analysis looking at brain volume differences over time between patients undergoing DMT and patients not under treatment at follow-up. There were no significant differences (Supplement Table 2). 

23. Table 3: not clear if presented p-values were corrected or not, if not, please provide corrected p-values (q-values).

We agree with the reviewer that this was not clear in Table 3. We have now changed the heading ‘p-value’ to ‘p-uncorrected’. At the bottom of the table we had previously explained that any p-value printed in bold had survived FDR correction and we hope that this has now become clearer. We don’t think it is necessary to provide q-values, as the original p-values are a better representation of the actual results taken from the data. We have also explained in the text that we have set our FDR threshold to 0.05, meaning that any p-value in bold was significant at this level.

24. Since the aim of the paper is to compare the volumetric and VBM approach, it would be helpful to provide a table/figure comparing the regions found by both approaches. It will be rather hard for the reader now to see the overlap between the two approaches since figure 3 does not include subcortical areas.

We agree with the reviewer and have now made a figure combining volumetric (regional GM, cerebellar GM and subcortical areas) and VBM results for easier comparison. We refer to the figure in results section 3.4.

“See Fig 5 for a visual comparison of volumetric (regional GM, cerebellar GM and subcortical areas) and VBM results.”

“Fig5. Comparison of voxel-based morphometry (VBM) and volumetric results in relapsing-remitting multiple sclerosis (RRMS). For illustration purposes, VBM results for significant grey matter (GM) change over time (w1-w0; pcorrected<0.001) (green) are overlaid on volumetric results (red) for regional GM, cerebellar GM and subcortical areas with significant volume decrease (w1-w0; q<0.05) (red). Areas where VBM and volumetry overlap are shown in yellow. Overlays are shown on an example subject’s axial (top row) and sagittal (bottom row) T1-weighted image. This figure was created using MRIcron (https://www.nitrc.org/projects/mricron).”

Discussion

25. Discussion is a bit generic but overall ok. The discussion could be improved by adding which atrophy rates you would expect for the different brain regions based on earlier studies and if the found atrophy rates are in line with that. Especially in terms of the relatively short follow-up of 1 year and early stage of the disease, reflect if your results are in line with literature.

We respectfully disagree with the reviewer that the discussion is generic, as we describe all observed results from both methods in detail and extensively compare them with the literature. The aim of our study was not to determine atrophy rates specifically, but rather observe areas where atrophy occurs. We agree with the reviewer that it would be interesting to determine regional atrophy rates, but suggest a different analysis must be performed, specifically aimed at determining annualised atrophy rates, and which should possibly involve an additional longer follow-up. This is beyond the scope of this paper, but may certainly be a possibility with future 5-year follow-up data from the same cohort, currently being collected.

26. What is cortical NAWM?

We thank the reviewer for identifying this mistake. We meant ‘cerebral NAWM’ here and by accident wrote ‘cortical’. The error has now been corrected.

27. It is important to discuss in more detail why you think the volumetric approach found more atrophic regions compared with the VBM approach. This is not expected, how do you interpret and justify this difference?

We agree with the reviewer that it is important to discuss why there are differences in results between the two approaches, and have addressed this in the discussion under section ‘4.4 Volumetry and VBM’. There could be multiple reasons for observing different findings using these two methods, as detailed in the discussion. We speculate, however, that one of the key contributors is that atrophy in MS is a widespread rather than a clustered process. This is reflected by a higher number of atrophic areas being detected using volumetry (detects any voxel loss) than VBM (places emphasis on clusters of change). We are unsure why the reviewer suggest the volumetric results are unexpected and would like to request further clarification. 

28. “Surprisingly, we also did not detect volumetric changes within areas in the frontal lobe associated with sensorimotor functioning.” Explain in more detail why you expected this, is the other early MS cohort that found this comparable to yours in terms of disease duration and disability?

We expected to observe volumetric changes within the sensorimotor cortex, as sensorimotor functioning is impaired in MS. We have now added this to the text for further clarification. 

“Surprisingly, we did not detect volumetric changes within areas in the frontal lobe associated with sensorimotor functioning, typically impaired in MS, such as in the precentral GM/NAWM and paracentral GM.”

29. Do you have any recommendations which method should be used/is best to assess regional atrophy? What are the main advantages/disadvantages of both methods?

Although we understand why the reviewer is interested in this, we do not think we are in a position to comment on these questions. We have only qualitatively compared the results of the different methods, as to understand why disparate results in the literature are evident. We have not performed any quantitative analyses, for example with ground truth segmentations, to be able to give a recommendation on which method should be used over the other, nor was this our intention. We have, however, extensively described the results and methods in the discussion to highlight the differences and advantages/disadvantages of both methods.

Conclusion

30. “Atrophy measures targeted to the most affected regions may provide more sensitive and specific markers of neurodegeneration, as imaging stratifiers of disease progression and endpoints for future therapeutic trials.” Regional measures being more sensitive biomarkers than global ones has not been discussed in the discussion and has not been tested in this study.

We agree with the reviewer and have rephrased this, as well as the corresponding sentence in the abstract, to better reflect the results of the study.

“Atrophy measures targeted to these regions may provide markers of neurodegeneration, and may be used as imaging stratifiers and endpoints for clinical decision making and therapeutic trials.”

31. “Analyses based on volumetry and VBM demonstrate different patterns of atrophy, albeit with some regional overlap.” Clarify which regions were most affected according to both analyses and what the differences where.

We agree with the reviewer that it is essential to discuss this and an extensive discussion of the results and the differences between the two methods can be found in the discussion of the manuscript. If there is anything the reviewer feels that was missed, we would kindly ask for further clarification.

---

## [Editor Report · Decision Letter 1]

10 Jul 2023

Patterns of brain atrophy in recently-diagnosed relapsing-remitting multiple sclerosis

PONE-D-22-30246R1

Dear Dr. Meijboom,

We’re pleased to inform you that your manuscript has been judged scientifically suitable for publication and will be formally accepted for publication once it meets all outstanding technical requirements.

Kind regards,

Niels Bergsland

Academic Editor

PLOS ONE
---

## [Editor Report · Acceptance letter]

21 Jul 2023

PONE-D-22-30246R1 

Patterns of brain atrophy in recently-diagnosed relapsing-remitting multiple sclerosis 

Dear Dr. Meijboom:

I'm pleased to inform you that your manuscript has been deemed suitable for publication in PLOS ONE. Congratulations! Your manuscript is now with our production department. 

Kind regards, 

on behalf of

Dr. Niels Bergsland 

Academic Editor

PLOS ONE